# ATP-dependent membrane remodeling links EHD1 functions to endocytic recycling

Raunaq Deo [1], Manish S. Kushwah [1], Sukrut C. Kamerkar[1], Nagesh Y. Kadam[2], Srishti Dar[1], Kavita Babu[2], Anand Srivastava [3] & Thomas J. Pucadyil [1]

Endocytic and recycling pathways generate cargo-laden transport carriers by membrane fission. Classical dynamins, which generate transport carriers during endocytosis, constrict and cause fission of membrane tubes in response to GTP hydrolysis. Relatively, less is known about the ATP-binding Eps15-homology domain-containing protein1 (EHD1), a dynamin family member that functions at the endocytic-recycling compartment. Here, we show using cross complementation assays in *C. elegans* that EHD1's membrane binding and ATP hydrolysis activities are necessary for endocytic recycling. Further, we show that ATP-bound EHD1 forms membrane-active scaffolds that bulge tubular model membranes. ATP hydrolysis promotes scaffold self-assembly, causing the bulge to extend and thin down intermediate regions on the tube. On tubes below 25 nm in radius, such thinning leads to scission. Molecular dynamics simulations corroborate this scission pathway. Deletion of N-terminal residues causes defects in stable scaffolding, scission and endocytic recycling. Thus, ATP hydrolysis-dependent membrane remodeling links EHD1 functions to endocytic recycling.

[1] Indian Institute of Science Education and Research, Dr. Homi Bhabha Road, Pashan, Pune 411008 Maharashtra, India. [2] Indian Institute of Science Education and Research, Sector 81, S.A.S Nagar, Mohali 140306 Punjab, India. [3] Molecular Biophysics Unit, Indian Institute of Science, Bangalore 560012 Karnataka, India. These authors contributed equally: Raunaq Deo, Manish S. Kushwah. Correspondence and requests for materials should be addressed to T.J.P. (email: pucadyil@iiserpune.ac.in)

Endocytosis and recycling pathways are vital to cellular physiology as it regulates nutrient uptake and display of adhesion molecules, ion-channels, and antigen-presenting receptors. Recycling is managed by the endocytic recycling compartment (ERC) or the tubular recycling endosome (TRE), which is a dynamic organelle comprised of a network of membrane tubules and vesicles concentrated in the perinuclear region[1-3]. The ERC receives a high density of soluble and membrane-bound cargo from endocytic vesicles, which are then sorted and released for recycling in transport carriers from this compartment. The mechanisms by which transport carriers are released from the ERC remain ill defined. A central player in ERC dynamics is the evolutionarily conserved EHD1 ATPase[4-6]. Mammals have 4 paralogs EHD1-4, which display ~70% amino acid identity. Despite such high sequence similarity, EHD proteins are distributed to diverse cellular compartments. EHD1 and 3 are localized predominantly to the ERC, EHD2 is present at the plasma membrane and EHD4 is localized to a Rab5-positive early endocytic compartment[7-9]. Studies in model organisms with a single ortholog of EHD, most similar to EHD1 in mammals, have revealed functions associated with endocytosis and recycling. The C. elegans ortholog, RME-1, facilitates recycling of internalized receptors from the ERC to the plasma membrane[10] and the ortholog in D. melanogaster Past1 is involved in endocytosis[11]. EHD1 knockout mice display phenotypes that range in severity from defects in spermatogenesis and infertility to embryonic lethality due to aberrant sonic hedgehog signaling and formation of primary cilia[12-14]. The depletion of EHD1 by RNAi elaborates the ERC into prominent membrane tubules, whereas the addition of purified EHD1 to semi-permeabilized cells leads to the loss-of-membrane tubules marked by the ERC-resident MICAL-L1 protein[15-17]. Such reciprocal phenotypes indicate that EHD1 participates in membrane remodeling and fission at the ERC. EHD proteins contain a dynamin-like, low-affinity ATP-binding G-domain, and self-assemble into electron-dense coats on tubulated liposomes, which in turn stimulate ATP hydrolysis[8,18]. These attributes would suggest that EHDs function similarly as classical dynamins in membrane remodeling and fission[8]. Despite these insights, the precise nature of and the role of ATP hydrolysis in EHD1's membrane remodeling functions remain uncharacterized.

Here, using cross-complementation assays in C. elegans, we find that EHD1's membrane binding and ATP hydrolysis activities are necessary for endocytic recycling. We then analyze the behavior of purified EHD1 in reconstitution assays on templates that mimic the tubular ERC topology. Our results indicate that a preference for high-positive membrane curvature directs ATP-bound EHD1 to bind and organize into membrane-active scaffolds that bulge the underlying tube. ATP hydrolysis promotes self-assembly of the scaffold, which propagates the bulge along the length of the tube causing intermediate regions to thin down and undergo fission. Coarse-grained molecular dynamics simulations of scaffolds of different lengths corroborate this pathway to fission. The N-terminal residues in EHD1 are important as their absence renders membrane bulges to catastrophically disappear in presence of ATP leading to defects in stable scaffolding, scission and endocytic recycling. Remarkably, we find that the closely related EHD2 shows a significantly lower ATPase activity and, concomitantly, a dramatically decreased membrane scission activity thus highlighting fundamental differences among EHD paralogs.

## Results

### Functional determinants required for endocytic recycling.
Previous analysis with mammalian cells that were depleted of or knocked out for EHD1 has revealed mild phenotypes of delayed cargo recycling from the ERC[16,19,20], likely due to compensation by the other orthologs. The single ortholog in C. elegans, RME-1 functions in recycling and null mutants display vacuoles in intestinal cells because of an expansion of the ERC caused by the block in vesicular transport out of this organelle. Remarkably, this phenotype can be rescued by the expression of human EHD1 thus highlighting an evolutionarily conserved function[10,20]. We used this assay as a facile screen to identify attributes in human EHD1 necessary to support endocytic recycling. Consistently, microinjection of Texas Red-labeled bovine serum albumin (TxRed-BSA) into the pseudocoelom showed no apparent phenotype in WT (N2 Bristol) worms but revealed numerous vacuoles in rme-1 (marked by asterix in Fig. 1a). Importantly, expression of the WT EHD1 in rme-1 significantly rescued this phenotype (Fig. 1b). Previous analysis of EHD2 has revealed residues required for membrane binding (F322), ATP-binding (T72), and ATP-hydrolysis (T94)[8]. We tested whether these mutants in the EHD1 background could complement the rme-1 phenotype. Expression of each of these mutants either showed complete absence of or substantially compromised rescue (Fig. 1a, b). Together, these results emphasize the role of membrane binding as well as ATP hydrolysis for EHD1 function. To understand how these biochemical traits relate to function, we tested purified EHD1 on compositionally simple membrane templates.

### Relationship between membrane binding and ATPase activity.
EHDs bind a diverse array of anionic phospholipids suggesting that their recruitment is facilitated by membrane charge and likely independent of stereo-specific recognition of lipid headgroups[7,8,16,21-23]. To minimize complexity, we started with membranes containing phosphatidylserine (PS) as the sole anionic lipid species. Importantly, previous reports indicate that PS recruits EHD1 to the ERC[16]. Co-sedimentation assays with EHD1 mixed with liposomes of progressively increasing PS content showed efficient membrane binding (Fig. 2a). Densitometric analysis revealed a sigmoidal rise in binding leading to saturation, with an $EC_{50}$ of 40 mol% PS (Fig. 2b), similar to that reported earlier[16]. Concomitantly, EHD1's basal ATP hydrolysis rate was stimulated in presence of liposomes with increasing PS content. Liposomes with 40% PS showed 16-fold stimulation in basal ATPase activity, reaching rates of 6 $min^{-1}$ (Fig. 2b). In contrast, the membrane-binding mutant F322A showed an $EC_{50}$ of 80 mol% PS but under these conditions showed stimulated ATPase activity (Fig. 2c). Thus membrane binding stimulates EHD1's ATPase activity, like is seen for dynamin[24,25]. Consistent with membrane charge being important, inclusion of 5 mol% phosphatidylinositol-4-phosphate (PI4P), an anionic lipid with higher charge density, lowered the PS requirement for membrane binding and stimulation of ATPase activity (Fig. 2d).

To determine whether the ATPase cycle in turn controls membrane recruitment, we assayed liposome co-sedimentation with the non-hydrolysable analog AMP-PNP, the slowly hydrolyzing analog ATP-γ-S and ADP. ATP-γ-S was hydrolyzed at a tenfold slower rate than ATP (Supplementary Figure 1), and was used instead of ATP to avoid significant consumption of the nucleotide within the time frame of liposome co-sedimentation assays. Under conditions of low (20 mol% PS) and moderate (40 mol% PS) charge, EHD1 bound membranes better with AMP-PNP than in the absence of nucleotides (Apo), which is consistent with recent FRET-based analysis with EHD2[26]. Remarkably, membrane binding was further enhanced with ATP-γ-S on liposomes (Fig. 2e). Binding in the presence of ADP was similar to that seen in the Apo state. Binding became

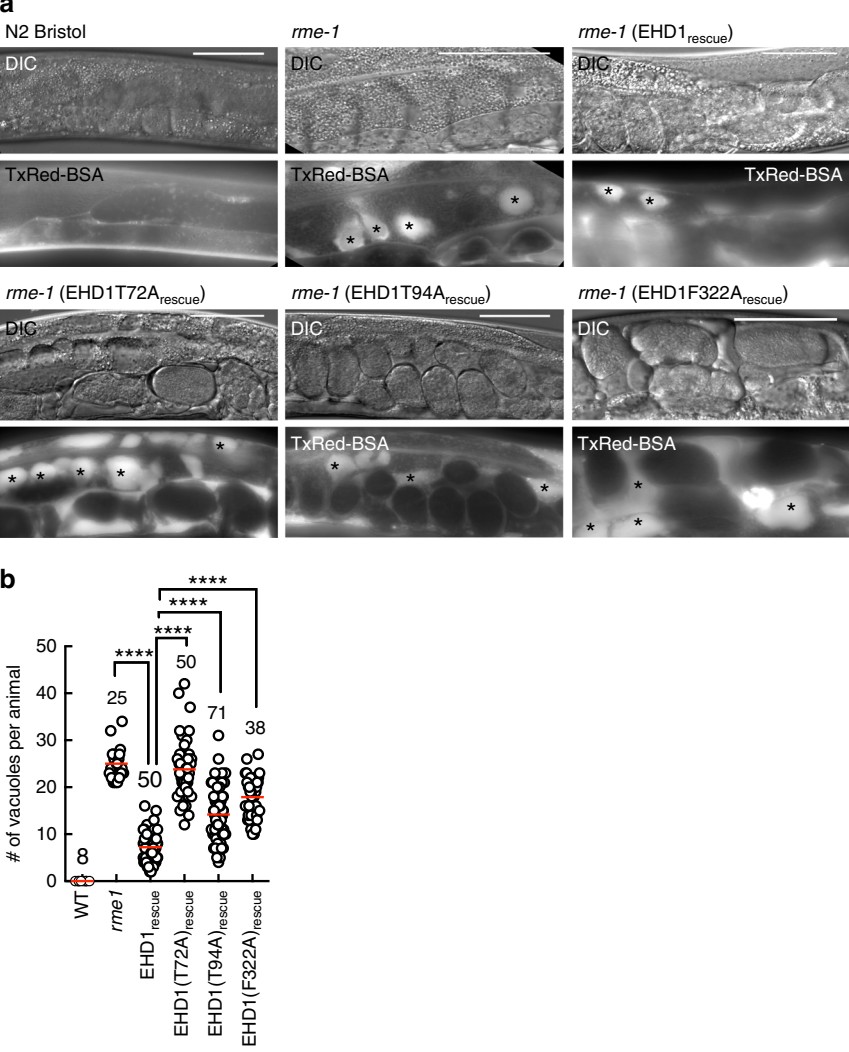

**Fig. 1** Functional determinants in EHD1 required for endocytic recycling. **a** Representative DIC and fluorescence images of worms injected with fluorescent BSA. Asterisks mark vacuoles. Scale bars = 50 μm. **b** Plot showing vacuole numbers per worm for the indicated numbers of worms. Red line denotes the mean. Statistical significance was assessed using an unpaired two-tailed *t*-test and ****P < 0.0001

nucleotide-independent on membranes of high (60 mol% PS) charge (Fig. 2e).

To visualize dynamics and distribution of EHD1 at discrete stages during its ATPase cycle, we turned to fluorescence microscopy on a template prepared of moderate (40 mol% PS) charge. These templates are formed by flow-induced extrusion of a membrane reservoir (Fig. 3a)[27,28], and display a planar supported lipid bilayer (SLB) that is connected to an array of unilamellar membrane tubes. Templates contained an environ-ment- and membrane curvature-insensitive fluorescent lipid, *p*-Texas Red DHPE for visualizing the membrane and estimating tube dimensions according to methods reported earlier[28]. Tubes in these templates display a wide range of sizes (Fig. 3b), which together with the SLB mimics the gradient of membrane curvature seen in the tubulovesicular ERC[29]. As a control, the PS-specific LactC2 domain[30] showed uniform distribution on both the SLB and tubes (Fig. 3c). To visualize EHD1 on these templates, we used a construct fused with EGFP at the C-terminus (EHD1-EGFP), which showed similar ATPase activity as WT (Supplementary Figure 1B). In contrast, N-terminal EGFP or GST fusions, used earlier to assess EHD function and distribution[4,8,15,31], displayed enhanced basal ATPase activity

and showed no stimulation with PS liposomes (Supplementary Figure 1B). The possible reason for the N-terminal fusions not behaving like WT could be due to disruption of an important oligomerization interface (see below). We therefore recommend caution to the use of such constructs in analyzing distribution and function of EHD proteins in cells.

Templates were incubated with EHD1-EGFP for 10 min, washed with buffer and imaged. These experiments revealed that the presence of AMP-PNP dramatically improved membrane binding, consistent with results from liposome co-sedimentation assays (Fig. 2e), but remarkably so did high membrane curvature. Thus, compared to a small 1.4-fold enrichment seen for the LactC2 domain, AMP-PNP-bound EHD1-EGFP showed eight-fold higher protein density on the curved membrane tubes than the planar SLB (Fig. 3d). This curvature preference was also seen on tubes of varying sizes (Fig. 3e). Binding with ADP was similar to that in the Apo state and under both these conditions, the curved membrane tube recruited more protein than the planar SLB (Fig. 3d). Furthermore, in contrast to the uniformly distributed LactC2 domain (Fig. 3c), AMP-PNP-bound EHD1 appeared organized as oligomers on the SLB (Fig. 4a, white arrowheads) and tubes (Fig. 4b, yellow arrowheads). The

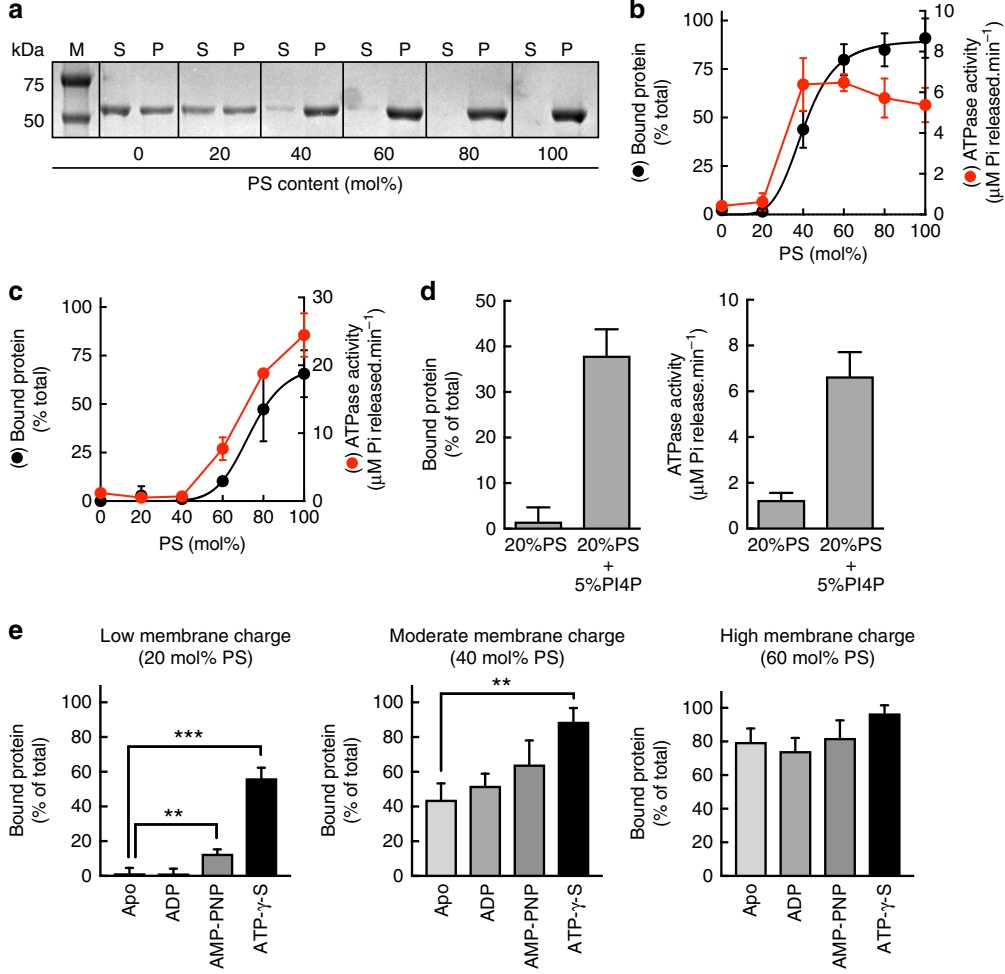

**Fig. 2** Membrane binding and ATPase activity of EHD1. **a** Representative gel showing results from a sedimentation assay with EHD1 added to liposomes of increasing PS content and spun down to separate the supernatant (S) and pellet (P) fractions. M denotes the marker lane. **b** Bound fraction and ATPase activity of EHD1 with liposomes of increasing PS content. Data represent mean ± SD for $N = 3$. **c** Bound fraction and ATPase activity of EHD1(F322A) with liposomes of increasing PS content. Data represent mean ± SD for $N = 3$. **d** Bound fraction and ATPase activity of EHD1 on liposomes of the indicated compositions. Data represent mean ± SD for $N = 3$. **e** Liposome-bound fraction of EHD1 with increasing PS with different nucleotides (mean ± SD, $N = 3$ for Apo, $N = 3$ for ADP, $N = 3$ for AMP-PNP, $N = 5$ for ATP-γ-S). Statistical significance was assessed using an unpaired two-tailed $t$-test and $**P = 0.003$, $***P = 0.0001$

presence of ATP-γ-S dramatically improved membrane recruitment of EHD1-EGFP (Fig. 4c), which is consistent with results from liposome co-sedimentation assays (Fig. 2e). Under these conditions, the protein appeared to be distributed as long filaments on the SLB and covered a significant area of the tubes (Fig. 4c). Flowing ATP to EHD1-EGFP assembled with ATP-γ-S lead to the rapid dissociation of the protein from the SLB (Fig. 4d). In comparison, dissociation from the tubes was significantly slower indicating that membrane curvature facilitates protein retention on the membrane in the wake of ATP hydrolysis (Fig. 4d). Together, we find that ATP-binding facilitates recruitment of EHD1 onto the membrane, whereas conversion to the ADP-bound state, likely due to assembly-stimulated ATP hydrolysis, causes membrane dissociation. These results are in agreement with cellular data with EHD2 where the ATP-binding defective T72A mutant displays a cytosolic distribution while the ATP hydrolysis defective T94A mutant appears membrane-localized[8]. Significantly, EHD1's preference for a curved membrane surface renders it resistant to ATP hydrolysis-induced membrane dissociation.

**Membrane remodeling**. Oligomers formed on tubes with EHD1-EGFP and AMP-PNP represent membrane-active scaffolds since tube fluorescence under them appeared brighter than adjacent regions (Fig. 5a). On tubes with EGFP selectively localized to the inner monolayer (Fig. 5b), addition of EHD1 with ATP-γ-S showed coincident changes in EGFP and membrane fluorescence thus confirming membrane remodeling and not a trivial protein-binding-induced change in membrane fluorescence. As brightness of a diffraction-limited membrane tube is proportional to the net membrane area occupied within the focal volume[32], bright regions on the tube under EHD1 scaffolds signify a membrane bulge. EHD1 scaffolds appeared well separated on a narrow tube and closely spaced on a wide tube (Supplementary Figure 2A, B). EHD1-EGFP showed similar effects on freestanding tubes thus ruling out possible artifacts caused by the surface attachment of tubes in the templates (Supplementary Figure 2C). In stark contrast, scaffolds of Alexa488-labeled dynamin1 coincided with regions of dim membrane fluorescence and signify membrane constriction (Fig. 5c), which is consistent with previous reports[27,33,34]. These differences are quantified in a Pearson's

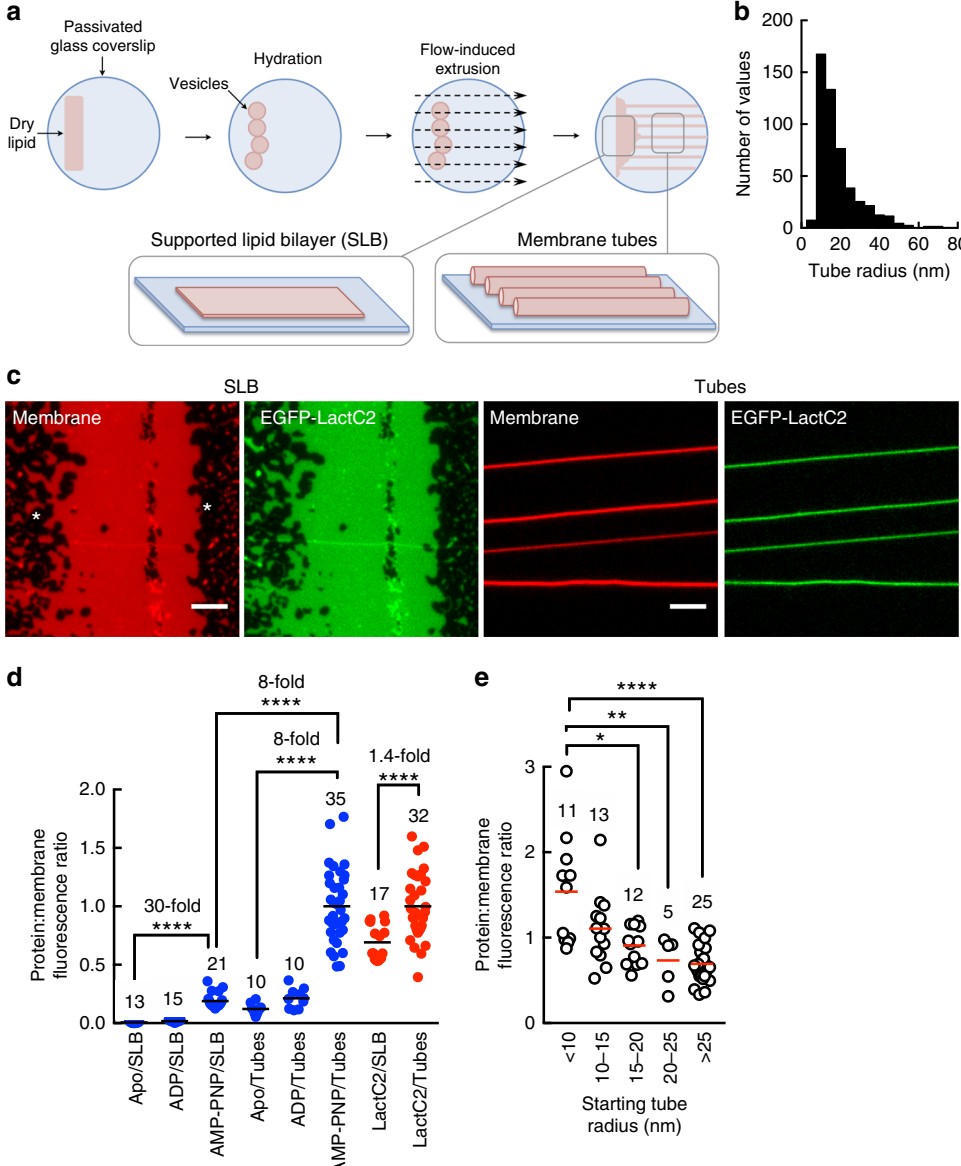

**Fig. 3** ATP- and curvature-sensitive membrane binding of EHD1. **a** Schematic of generation of membrane templates used in the study. **b** Distribution of tube radii from five independent preparations of templates. **c** Representative fluorescence micrographs showing distribution of EGFP-LactC2 on SLBs and tubes. Asteriks mark the bare glass surface. Scale bars = 5 μm. **d** Protein:membrane fluorescence ratios of EHD1-EGFP with various nucleotides (blue) and GFP-LactC2 (red) for the indicated numbers of tubes and SLBs sampled. Black lines denote means. Statistical significance was assessed using Mann–Whitney test and ****$P < 0.0001$. **e** Protein:membrane fluorescence ratios of AMP-PNP-bound EHD1-EGFP on tubes of varying starting sizes for the indicated numbers of tubes. Red lines denote means. Statistical significance was assessed using Mann–Whitney's two-tailed test and *$P = 0.010$, **$P = 0.005$, ****$P < 0.0001$

analysis correlating protein to membrane fluorescence, where AMP-PNP-bound EHD1 showed a positive correlation, whereas dynamin showed a negative correlation (Fig. 5d). Furthermore, the bulged tube radius ($R_b$) under AMP-PNP-bound EHD1 scaffolds varied with the starting tube size (Fig. 5e), although the constricted tube radius ($R_c$) under dynamin scaffolds remained constant at ~11 nm (Fig. 5f), an estimate that agrees well with previous values reported from force-spectroscopic and cryoEM analyses thus validating our approach of estimating tube dimensions[33,35]. Such plasticity in membrane remodeling of EHD1 is consistent with the loosely organized EHD coats seen on tubulated liposomes under EM[8,18,36], quite unlike the rigid and highly ordered dynamin scaffold that constricts the underlying tube to a well-defined

geometry[35]. Thus, from the standpoint of remodeled membrane intermediates and mechanical attributes of the scaffold, EHD1 and dynamin are fundamentally different. Remarkably, membrane bulging by EHD1 scaffolds also remodeled intervening regions on the same tube. Thus, on a tube of 10 nm radius, bulging (Fig. 5g, green arrowheads) caused thinning of the intervening regions of the tube (Fig. 5g, blue lines), reaching estimates of ~7 nm in radius. Similar effects are seen with dynamin where a constriction causes bulging of adjacent regions[34]. Such reciprocal shape change in membranes arises from the necessity to conserve membrane area, which should imply that the tubes in our templates represent a system of finite reservoir. Despite such membrane remodeling, lipids exhibit free diffusion on tubes displaying EHD1 oligomers and dynamin

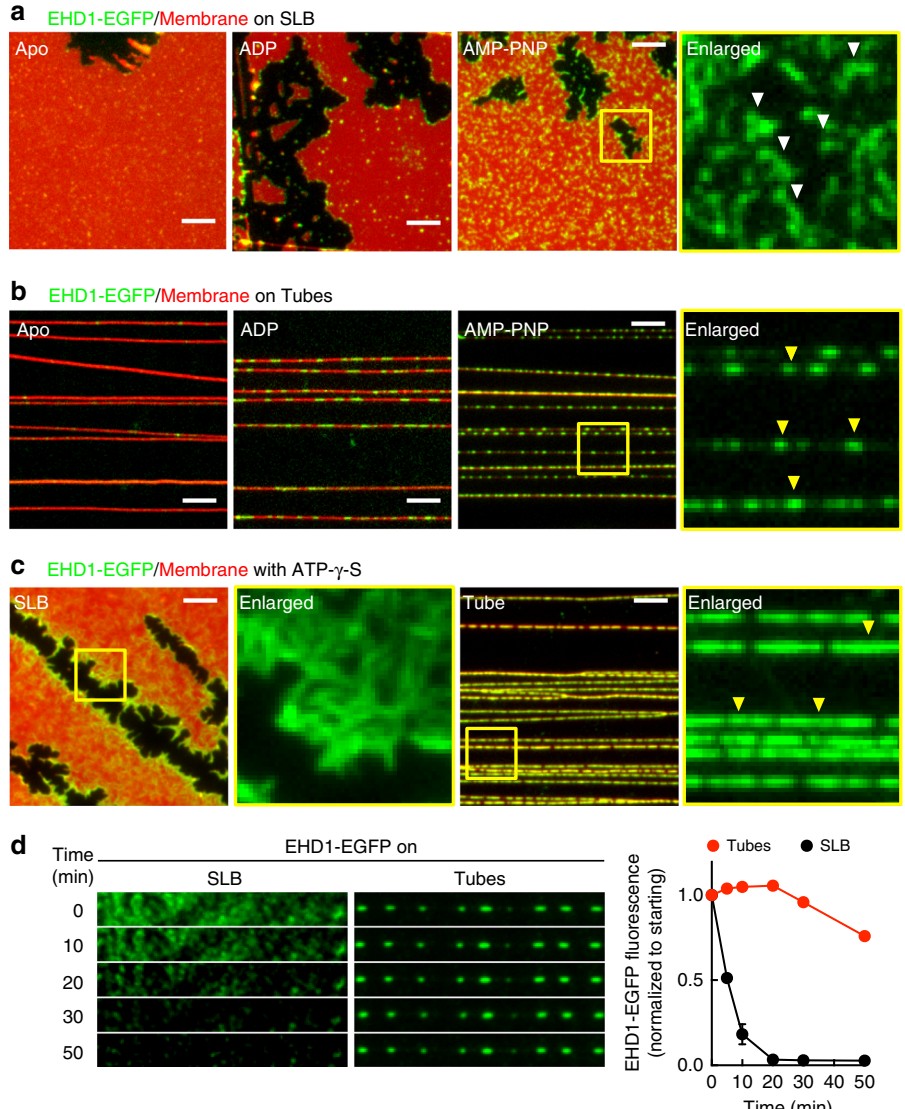

**Fig. 4** EHD1 distribution and dynamics on membranes. Representative fluorescence micrographs showing the distribution of EHD1-EGFP in the Apo, ADP-bound, and AMP-PNP-bound states on SLBs (**a**) and tubes (**b**). White and yellow arrowheads mark oligomers of EHD1-EGFP on the SLB and tubes, respectively. Scale bars = 5 μm. **c** Representative fluorescence micrographs showing the distribution of EHD1-EGFP with ATP-γ-S on the SLB and tubes. Yellow arrowheads mark oligomers of EHD1-EGFP on the tubes. Scale bars = 5 μm. **d** Representative fluorescence micrographs showing the effect of ATP addition to EHD1-EGFP assembled with ATP-γ-S on the SLB and tubes. Plot shows quantitation of fluorescence intensities of EHD1-EGFP on the SLB and tubes. Data represent mean ± SD of $n = 3$ ROIs on the SLB and tubes

scaffolds (Supplementary Figure 2D, E), which sets them apart from the Bin-Rvs-Amphiphysin (BAR) domain-containing proteins that impose a lipid-diffusion barrier[37,38].

Remarkably, flowing in EHD1 with ATP caused pronounced and rapid membrane bulging and associated thinning, which led to a significant fraction of the tubes undergoing scission (Fig. 6a, red arrowheads, see Supplementary Movie 1). The number of cuts on a single tube was low, possibly because the first cut led to sudden loss of tension (see Supplementary Movie 1). Also, not all tubes underwent scission (Fig. 6a, white arrowheads) and a systematic analyses revealed that membrane remodeling or bulging was apparent on tubes below 25 nm in radius. Within this range, the probability of bulging leading to tube scission was sensitive to the starting tube size (Fig. 6b). Scission was also seen on freestanding tubes and at a faster rate, perhaps due to the higher membrane tension (Supplementary Figure 3 and Supplementary Movie 2). However, analyzing membrane intermediates

was difficult due to out-of-focus movement of the tube. The membrane-binding mutant F322A, which showed an EC$_{50}$ of 80 mol% PS in membrane binding (Fig. 2c), showed no such effects on 40 mol% PS-containing tubes but remodeled and cut 80 mol% PS-containing tubes (Supplementary Figure 4).

Monitoring dynamics of EHD1-EGFP flowed on to templates with ATP revealed rapid self-assembly of scaffolds along the length of the tube (Fig. 6c, white arrowheads, Supplementary Movie 3). At tenfold lower protein concentrations of 0.1 μM (to reduce background fluorescence), scaffolds grew at an apparent rate of $27 \pm 5$ nm s$^{-1}$ (mean ± S.D., $n = 5$). This in turn led to the bulged region on the tube (Fig. 6d, white arrowheads) to expand causing intervening regions to thin down (Fig. 6d, green arrowheads). Thus, reactions with ATP soon led to significant coverage of the tube with EHD1. On a tube of 10 nm radius, self-assembly caused intervening regions to thin down to ~5 nm radius (Fig. 6e), before undergoing scission. The estimate of 5 nm

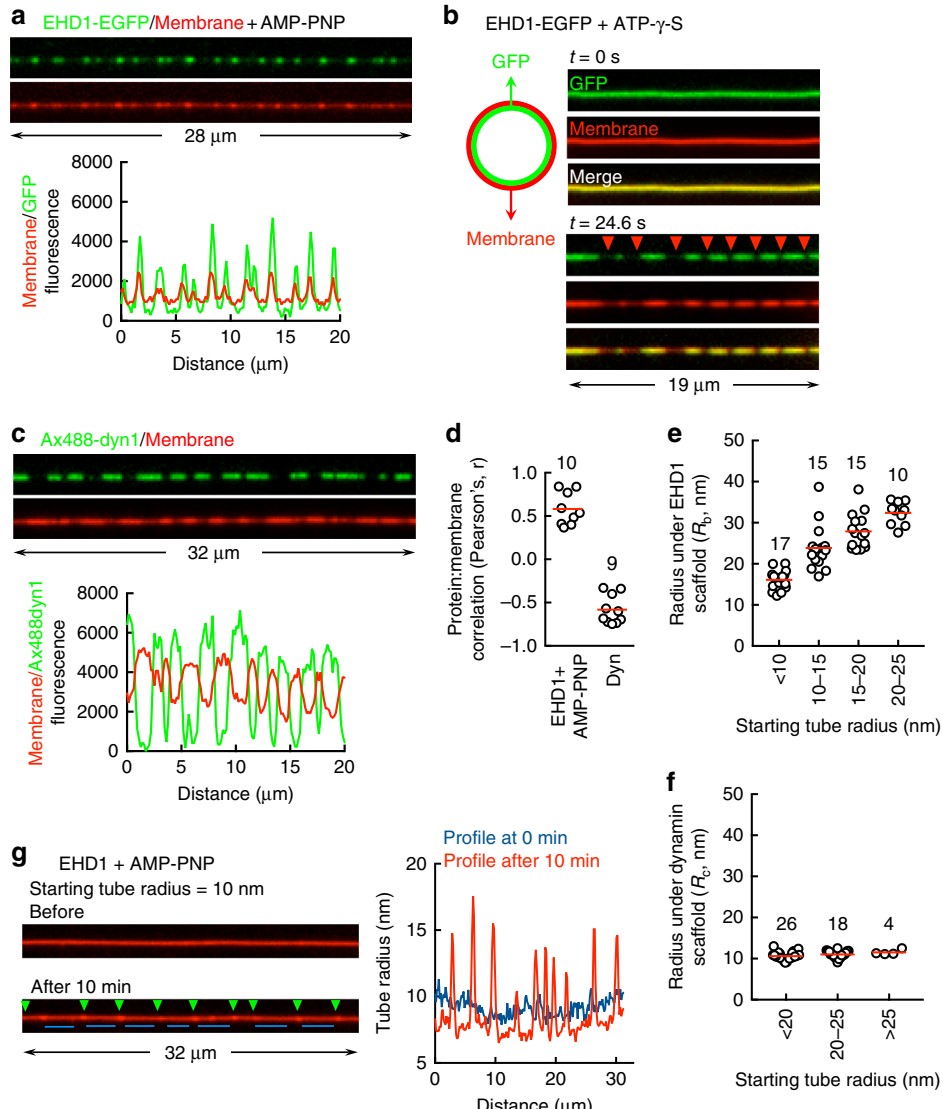

**Fig. 5** Membrane remodeling by EHD1. **a** Representative images and fluorescence line profiles of tubes showing scaffolds of AMP-PNP-bound EHD1-EGFP. **b** Representative images showing coincidence between GFP and membrane fluorescence in response to formation of extended EHD1 scaffolds with ATP-γ-S. Red arrowheads mark sites of tube thinning. **c** Representative images and fluorescence line profiles of tubes showing Alexa488-labeled dynamin1 scaffolds. **d** Pearson's correlation between protein and membrane fluorescence for the indicated numbers of tubes. Red lines denote means. **e** Bulged tube radius ($R_b$) under AMP-PNP-bound EHD1 scaffolds as a function of the starting tube radius for the indicated numbers of tubes. Red lines denote means. **f** Constricted tube radius ($R_c$) under dynamin scaffolds as a function of the starting tube radius for the indicated numbers of tubes. Red line denotes mean. **g** Representative images and associated line profiles acquired in the membrane channel showing remodeling of a tube by AMP-PNP-bound EHD1 scaffolds. Green arrowheads mark bulged regions and blue lines mark thinned-down regions on the tube

is equivalent to the bilayer thickness and matches the critical requirement for membrane fission[39,40]. Consistent with such a pathway, kymographs of multiple single events revealed that the site of scission (Supplementary Figure 5, red arrowheads) was located between membrane bulges formed by EHD1 scaffolds (Supplementary Figure 5, black arrowheads). Thus, the unique ability of membrane bulging combined with an ATPase-dependent self-assembly appears sufficient for EHD1 to cause scission.

**Intermediates in the membrane-remodeling pathway.** Flowing EHD1 with AMP-PNP, ATP-γ-S, and ATP onto tubes and monitoring early stages of the reaction showed small but significant differences in the density of membrane bulges (Fig. 7a). As assembly stimulated ATP hydrolysis would be minimal at early stages and bulging should arise from the nucleation of EHD1 on the membrane, such differences in nucleation density could reflect different self-assembling properties of EHD1 with these nucleotides. This could explain differences in membrane binding and organization of EHD1 seen with AMP-PNP and ATP-γ-S (Figs. 2e, 4a, b). Despite these differences, ATP hydrolysis appeared to facilitate self-assembly and is evident from analysis with ATP-γ-S and the T94A mutant. Thus, reactions with ATP-γ-S, which was hydrolyzed at a tenfold slower rate than ATP (Supplementary Figure 1A), showed a slower rate of fission (Fig. 7b). These reactions were characterized by a longer time interval between the onset of membrane bulging and tube scission (defined as the fission time, Fig. 7c). Similar effects are also seen with the T94A mutant, which self-assembled as wild type to bulge the membrane (Supplementary Figure 6A), but on account of a

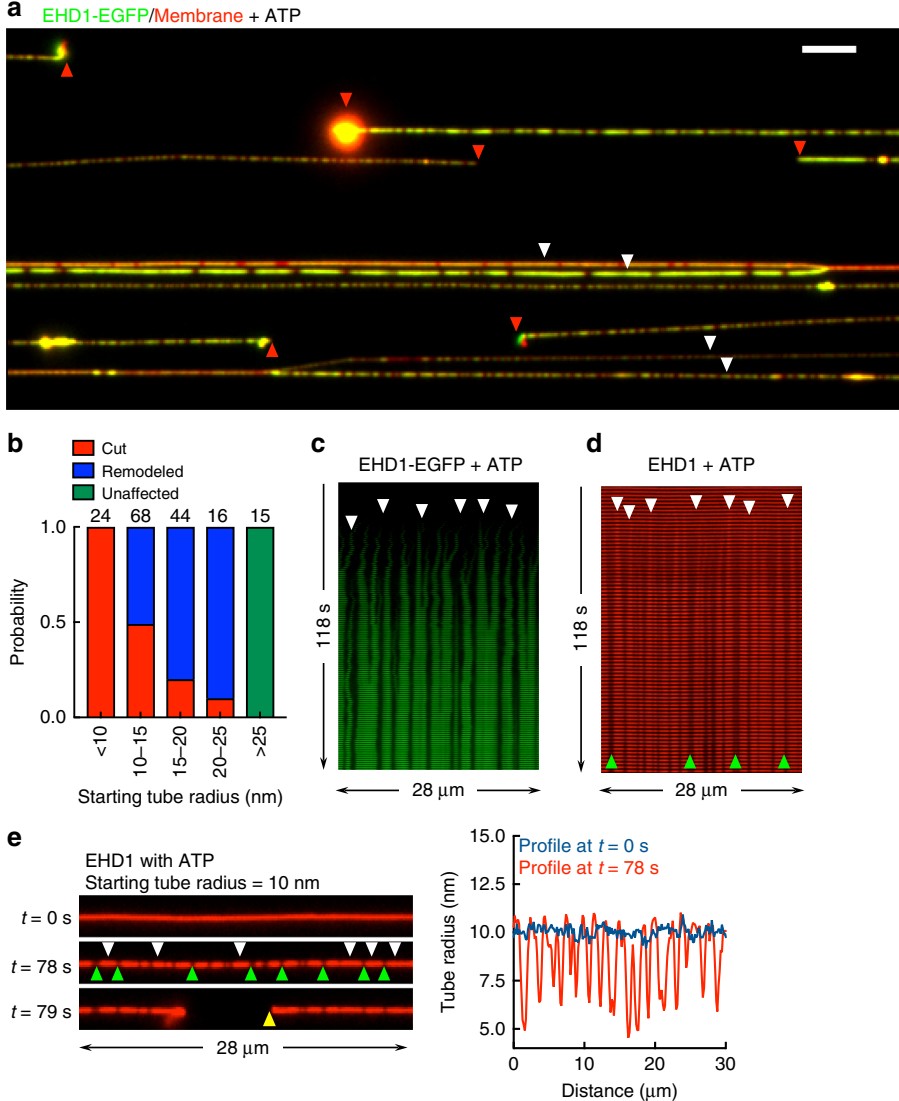

**Fig. 6** Membrane fission induced by ATP hydrolysis. **a** Representative fluorescence micrograph after 5 min of adding EHD1-EGFP with ATP to templates (see Supplementary Movie 1). Red arrowheads mark cut ends of tubes and white arrowheads mark tubes that remain uncut. Scale bar = 10 μm. **b** Tube remodeling and fission probability as a function of starting tube radius for the indicated numbers of tubes. **c** Representative montage of tube images acquired in the GFP channel showing the nucleation (white arrowheads) and growth of EHD1-EGFP scaffolds in presence of ATP (see Supplementary Movie 3). **d** Representative montage of tube images acquired in the membrane channel showing the expansion of bulges (white arrowheads). White arrowheads mark the site of onset of bulging and green arrowheads mark sites of tube thinning. **e** Frames and associated line profiles from a representative movie acquired in the membrane channel showing the bulged (white arrowheads) and thinned-down (green arrowheads) sites on a tube, before fission (yellow arrowheads)

tenfold slower rate of ATP hydrolysis (Supplementary Figure 1A), showed a slower rate of fission (Supplementary Figure 6B, C). Thus, slower ATP hydrolysis causes slower self-assembly, which in turn causes a delay in the formation of highly thinned-down regions in between scaffolds that undergo fission.

The slow kinetics of tube scission with ATP-γ-S allowed us to analyze intermediates by a quick washout of excess protein. Under these conditions, STED microscopy revealed faintly helical patterns in the EHD1-EGFP scaffolds (Fig. 7d). The tendency to form such helical scaffolds could explain EHD1's preference for positive membrane curvature (Fig. 3d). Lipid-diffusion measurements on tubes displaying extended scaffolds and highly thinned-down (but not cut) intervening regions revealed a significant ~50% drop in the mobile fraction (Fig. 7e, f), quite unlike the

unhindered lipid diffusion seen earlier with AMP-PNP where tube thinning in the intervening regions was shallow (Supplementary Figure 2D). Remarkably, flowing in micelles of octadecyl rhodamine (R18) showed that their fusion caused the diffusion-based spreading of R18 across multiple membrane bulges (Fig. 7g). As the externally added R18 diffuses in the outer leaflet[41], whereas FRAP samples lipid-diffusion in both leaflets, a mobile fraction of 50% points to a stalk-like hemifusion intermediate where the outer leaflet is continuous but the inner leaflet is discontinuous (schematized in Fig. 7h), like has been reported for dynamin[34,42]. Thus, despite the fundamentally different pathways by which they execute fission, both EHD1 and dynamin force the tube to visit similar pre-fission membrane intermediates.

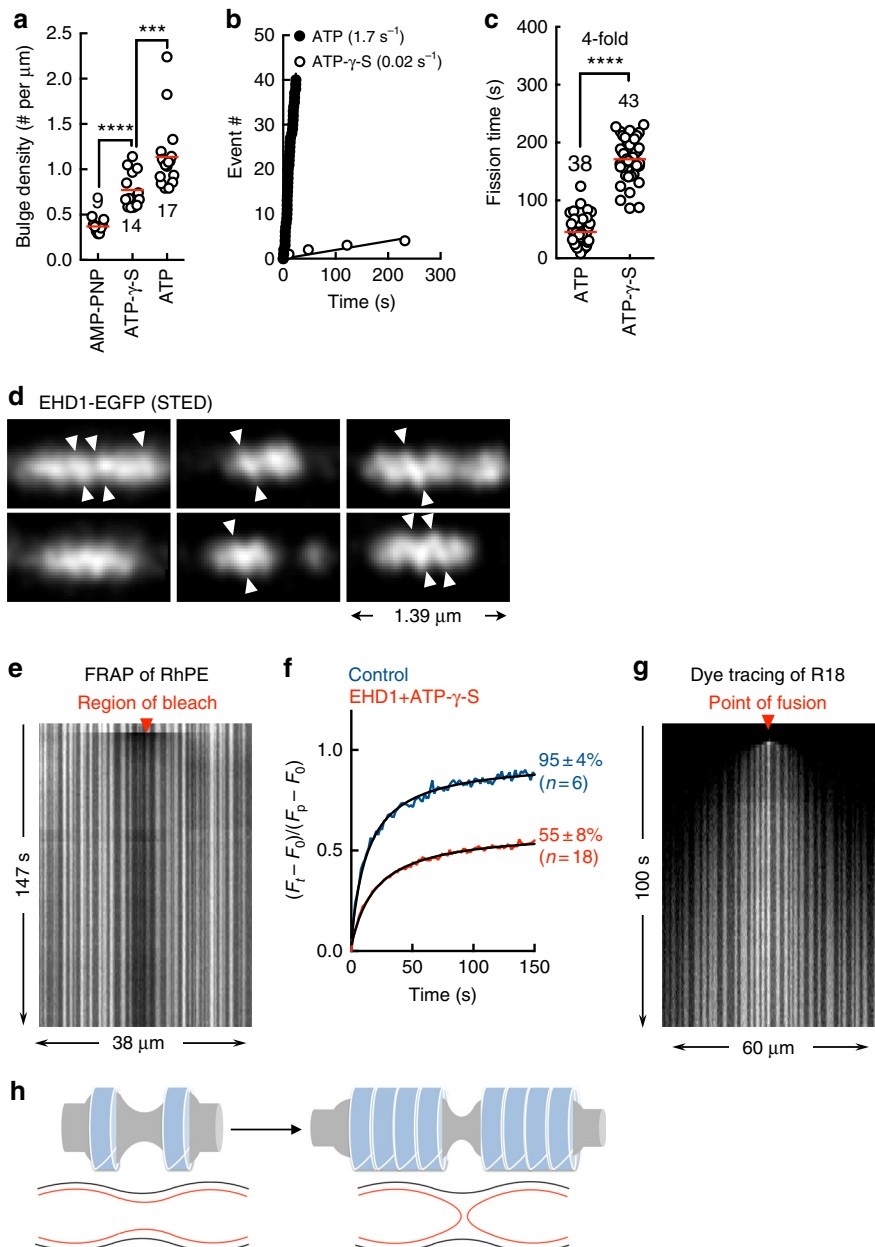

**Fig. 7** Intermediates in the membrane-remodeling pathway. **a** Bulge density per unit length of the tube upon addition of EHD1 with AMP-PNP, ATP-γ-S and ATP for the indicated numbers of tubes. Red line denotes the mean. Statistical significance was assessed using Mann–Whitney's two-tailed test and ***P < 0.005, ****P < 0.0001. **b** Plot showing cumulative cuts seen as a function of time (fission rate) upon adding EHD1 with ATP or ATP-γ-S to a collection of tubes. **c** Plot showing the time interval between membrane bulging and scission (fission time) upon adding EHD1 with ATP or ATP-γ-S to tubes for the indicated numbers of events. Red line denotes the mean. Statistical significance was assessed using Mann–Whitney's two-tailed test and ****P < 0.0001. **d** STED microscopic images of EHD1-EGFP scaffolds with ATP-γ-S. White arrowheads mark the possible orientation of helical rungs. **e** Representative kymograph showing FRAP of an RhPE-containing tube displaying EHD1 scaffolds with ATP-γ-S. **f** Fluorescence recovery kinetics of RhPE for the indicated conditions. Numbers represent mobile fraction (mean ± SD for n(tubes) as indicated). **g** Representative kymograph showing fusion and spreading of R18 on a tube displaying EHD1 scaffolds with ATP-γ-S. **h** Schematic showing growth-induced tube thinning leading to the formation of a hemifused intermediate. Black and red lines depict the outer and inner monolayers, respectively of the tube

**Simulations of scaffold-induced tube bulging and fission**. We performed coarse-grained molecular dynamics simulations to gain molecular insights into how a membrane tube would behave in response to an externally applied attractive scaffold that is larger in diameter than the starting tube. To mimic a short EHD1 scaffold, we placed a 4 nm-long scaffold on a 100 nm-long tube. Simulating this scenario shows that the tube under the scaffold becomes bulged (Fig. 8a). Interestingly, while the bulge extends to regions outside of the scaffold (Fig. 8b), the entire

system remains stable with time (Supplementary Movie 4). Performing simulations with a 20 nm-long scaffold (Fig. 8c), which should mimic the scenario reached upon adding EHD1 with ATP, produces a noticeably different outcome. The extent of tube bulging in this case is heighted which, on account of compensation of membrane area, causes progressive thinning of the tube at regions outside the scaffold (Fig. 8d, Supplementary Movie 5). Upon reaching a critical ~2 nm radius, lipids that are diametrically opposite in the lumen-facing leaflet are forced into close

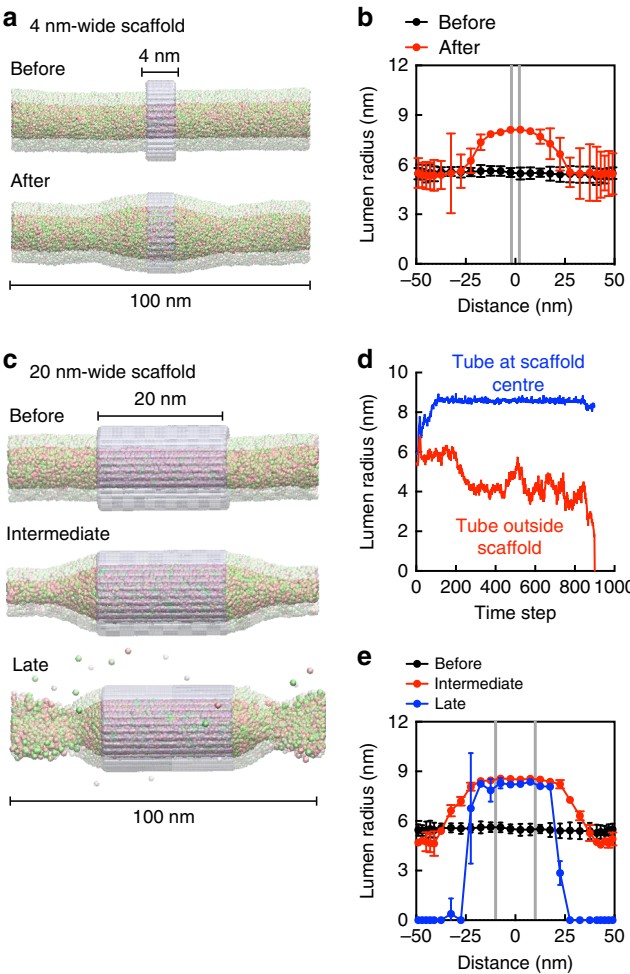

**Fig. 8** Molecular dynamics simulations of scaffold-induced tube bulging and fission. **a** Frames from simulations of a 4 nm-long scaffold (magenta) on a 100 nm-long membrane tube. Cream and green spheres represent headgroups of DOPS and DOPC lipids, respectively. Light and dark spheres represent headgroups of lipids on the outer and inner leaflet, respectively. Only lipid head groups are shown for clarity (see Supplementary Movie 4). **b** Lumen radius along the tube length upon placement of a 4 nm-long scaffold. Data represent the mean with variance. Gray lines represent scaffold boundaries. **c** Frames from simulations of a 20 nm-long scaffold on a 100 nm-long membrane tube. Cream and green spheres represent headgroups of DOPS and DOPC lipids, respectively. Light and dark spheres represent headgroups of lipids on the outer and inner leaflet, respectively. Only lipid head groups are shown for clarity (see Supplementary Movie 5). **d** Evolution of lumen radius at the middle (blue) and at the site of maximum thinning (red) after placement of a 20 nm-long scaffold. X-axis is in units of 10,000 τ, where τ is a CG time step. **e** Lumen radius along the tube length upon placement of a 20 nm-long scaffold. Data represent the mean with variance. Gray lines represent scaffold boundaries

proximity, leading to spontaneous fission (Fig. 8e and Supplementary Movie 5). At this stage, fluctuations in the lumen radius show a drastic increase, which suggest a possible mechanism for interleaflet mixing of lipids (Supplementary Movie 6). Of note, constraints of scaffold rigidity imposed in these simulations prevent the cut ends of the tube from resealing. Indeed, simulations of a bare tube with a lumen size of ~2–4 nm, like is reached at the end of simulations with a 20 nm-long scaffold, showed stochastic fission that is followed by the resealing and separation of the cut ends of the tube. Together, results from these

simulations corroborate the fission pathway observed with EHD1 in presence of ATP.

**Structural determinants for stable membrane remodeling.** Recent studies propose that the N-terminal region of EHDs facilitate an allosteric transition from a closed auto-inhibited state in solution to an open active conformation on the membrane. Thus, whereas the crystal structure of full-length EHD2 reveals a compact scissor-shaped dimer where the N-terminus is tucked in the G-domain, the structure of EHD4 deleted in N-terminal residues 1–22 reveals a more open conformation[43]. In addition, EPR spectroscopy with liposomes indicates that residues in the N-terminal region of EHD2 bind and partition into the membrane[36]. As the N-terminal region is partially conserved in all EHDs (Fig. 9a), we analyzed its functions in EHD1 by generating a mutant with a smaller deletion of residues 2–9, hereafter referred to as EHD1(Δ2–9). In cross complementation assays, EHD1(Δ2–9) was unable to rescue the *rme-1* phenotype indicating a significant defect in endocytic recycling (Fig. 9b, c). Liposome co-sedimentation assays revealed EHD1(Δ2–9) to be similar to WT in membrane binding properties (Fig. 9d). In addition, both the basal and assembly-stimulated ATPase activities of EHD1(Δ2–9) were similar to WT (Fig. 9e). Thus, if the N-terminal region was an allosteric regulator of ATPase activity, then EHD1(Δ2–9) should have shown higher basal ATPase activity, which is not the case. Remarkably, reactions monitoring EHD1(Δ2–9) on tubes with ATP showed less prominent bulging and no fission (Fig. 9f, Supplementary Movie 5), again establishing that efficient endocytic recycling requires membrane remodeling and fission by EHD1. Closer analyses of reactions with ATP monitored in the membrane fluorescence channel revealed the formation, but no expansion of the membrane bulge. Instead, bulges quickly dissipated and reform at the same location on the tube (Fig. 9g, white arrowheads). Together, these results indicate that the fission defect seen with EHD1(Δ2–9) arises from instability in the scaffold that disallows sustained self-assembly in response to ATP hydrolysis.

**Differences between EHD1 and EHD2.** Previous studies indicate that liposomes incubated with EHD2 and ATP undergo tubulation with a loosely ordered electron-dense coat on the membrane[8], which is quite unlike the extensive membrane remodeling and scission seen here with EHD1. To test whether this discrepancy reflects differences in assay conditions (preformed tubes versus liposomes), we analyzed EHD2's membrane binding, ATPase activity and membrane remodeling functions. In liposome co-sedimentation assays, EHD2 displayed a dramatically higher affinity for PS-containing liposomes, with half maximal binding seen at 10 mol% PS (Fig. 10a). Remarkably, EHD2 showed significantly lower ATPase activity than EHD1 (Fig. 10b). Thus, in presence of 100 mol% PS liposomes, EHD2 displayed an ATPase activity of 0.13 min$^{-1}$, which is consistent with earlier estimates[8], but is 40-fold lower than seen with EHD1. Consistently, EHD2 showed a significant delay in membrane remodeling with ATP, with bulges appearing only after long incubation, and drastically reduced scission efficiency (Fig. 10c, d). Together, our results comparing these proteins under identical experimental conditions highlight fundamental differences between the two paralogs.

**Discussion**
Our results with EHDs unravel an atypical mechanism by which a protein scaffold manages membrane remodeling. Thus, while the BAR family of proteins tubulate and dynamins constrict the membrane[44–46], EHDs cause membrane bulging. The

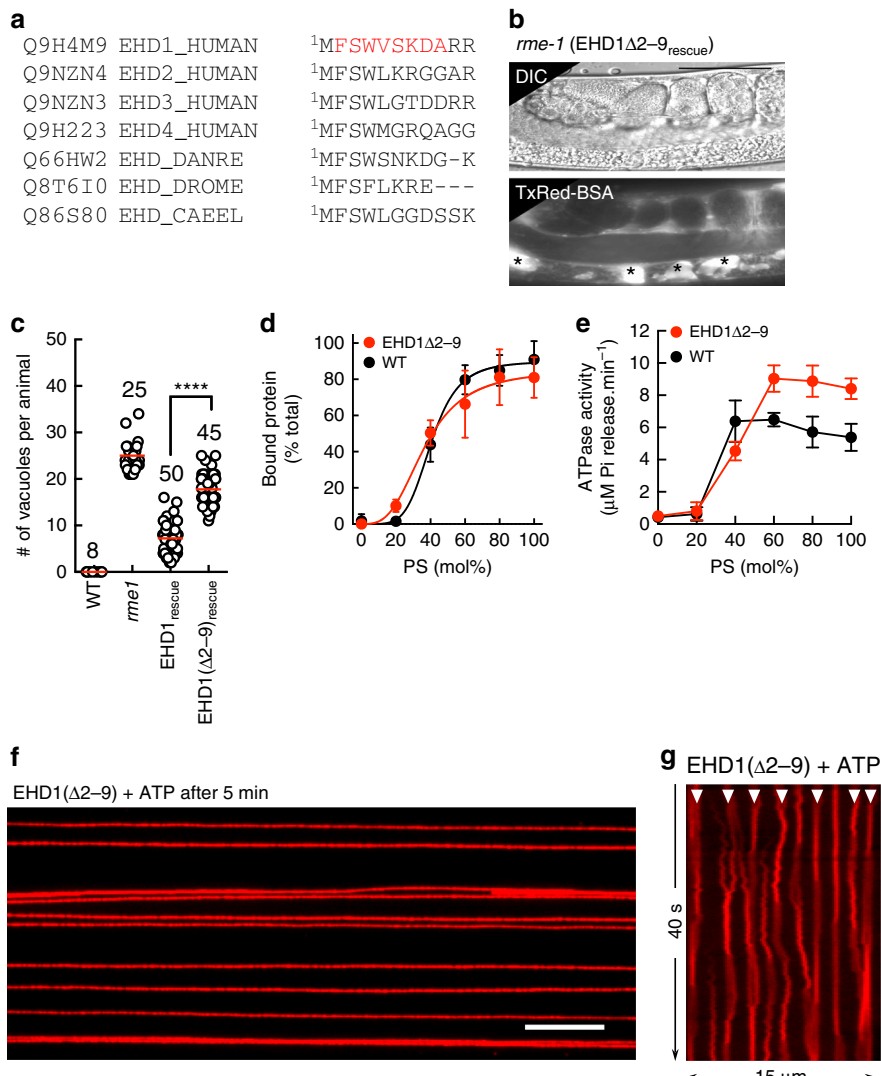

**Fig. 9** N-terminal residues are critical for EHD1 function. **a** Alignment of the N-terminal region in EHD proteins. **b** Representative DIC and fluorescence images of worms injected with fluorescent BSA. Asterisks mark vacuoles. Scale bar = 50 μm. **c** Plot showing vacuole numbers per worm for the indicated numbers of worms. Red line denotes mean. Data for WT, *rme-1* and EHD1$_{rescue}$ are shown for comparison. Statistical significance was assessed using an unpaired two-tailed *t*-test and ****$P < 0.0001$. **d** Bound fraction of EHD1(Δ2–9) with liposomes of increasing PS content (mean ± SD, $n = 3$). Data for WT is shown for comparison. **e** ATPase activity of EHD1(Δ2–9) with liposomes of increasing PS content (mean ± SD, $n = 3$). Data for WT is shown for comparison. **f** Representative micrograph showing EHD1(Δ2–9) added to templates with ATP after 5 min (see Supplementary Movie 6). Scale bar = 10 μm. **g** Representative kymograph of tube images acquired in the membrane fluorescence channel showing the formation and dissolution of membrane bulges (white arrowheads)

scissor-shaped EHD dimer displays a convex arc-shaped membrane binding surface[8,36]. We speculate bulging to be a consequence of the underlying lipid bilayer adapting to such a surface and further studies will be necessary to dissect the molecular basis of this unique form of membrane remodeling. The GTPase cycle in dynamins elicits conformational changes in the scaffold in order to impose further constriction. In contrast, ATP hydrolysis facilitates EHD1 self-assembly, which could arise from the stabilization of inter-rung interactions by the transition state, as is seen with dynamin[47,48]. On longer time scales, ATP hydrolysis causes membrane dissociation, possibly through a disassembly process that lowers the avidity of EHD-membrane interactions. Membrane dissociation is however countered by the propensity for EHD scaffolds to stably engage with a curved membrane surface. A predicted outcome is that at steady state, EHDs would be absent on relatively planar membranes but retained on membrane tubes, perhaps in a quasi-assembled state. The

N-terminal residues, possibly due to their tendency to partition into the membrane, confer stability to the scaffold in the wake of ATPase-induced disassembly. In their absence, EHD1 can nucleate to form a scaffold that bulges the tube but the scaffold displays catastrophic disassembly and dissolution in presence of ATP. On the other hand, due to its higher affinity for negatively charged membranes and slower ATPase activity, EHD2 may function to uniformly coat the membrane rather than self-assembling at discrete sites to induce membrane curvature. Consistently, loss of EHD2 has been shown not to cause a defect in fission and release of caveolae but to enhance their mobility on the membrane[49,50].

Our work combining in vitro reconstitution and molecular dynamics simulations reveal an intrinsic ability for EHD1 to remodel and cause membrane scission. Since the total membrane area on the tube should be conserved, bulging caused by the self-assembly of an EHD scaffold causes thinning of adjacent regions

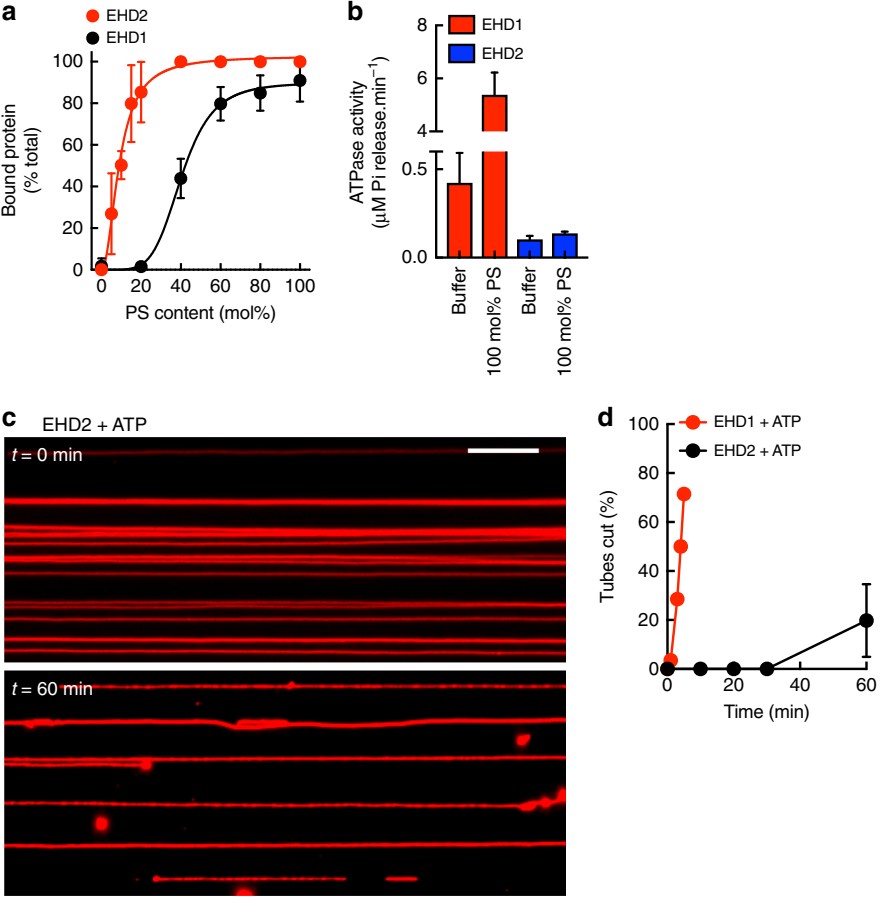

**Fig. 10** Functional differences between EHD2 and EHD1. **a** Bound fraction of EHD2 with liposomes of increasing PS content (mean ± SD, $N = 3$). Data for EHD1 are shown for comparison. **b** ATPase activity of EHD2 with and without PS liposomes (mean ± SD, $N = 3$). Data for EHD1 are shown for comparison. **c** Frames from representative movie showing EHD2 added to templates with ATP acquired at the indicated time periods. Scale bar = 10 μm. **d** Plot showing the number of tubes cut (as percentage of the total tubes analyzed) with the indicated proteins. Data represent the mean ± SD

on the same tube, which results in fission. Together, these results define an atypical mechanism for membrane fission. While these attributes are apparent on compositionally simple model membranes and it remains to be seen if complex native membranes, like the ERC tubules, respond similarly to EHD1, our results showing a correlation between membrane remodeling and fission to successful endocytic recycling strongly point to such a possibility. Importantly, previous results indicating that an absence of EHD1 leads to the expansion of ERC tubules supports the notion that it participates in membrane remodeling leading to fission at the ERC[16,17].

Vesicular transport pathways function to sort membrane-bound cargo and package them into transport carriers. For such a process to be effective, cargo sorting, and fission of transport carriers should display a strict temporal hierarchy. Models proposed for clathrin-mediated endocytosis indicate that while the early events of cargo sorting are managed by clathrin-dependent clustering of adaptor-bound cargo, an orchestrated build-up of binding partners and an increase in membrane curvature at late stages lead to the transient recruitment of dynamin at the necks of clathrin-coated pits, where it catalyzes fission[51]. On the other hand, at steady state, the ERC is abundant in lipid- and protein-binding partners of EHD1[16,21,52]. How then can EHD1's membrane remodeling functions be timed to follow cargo sorting? Recent work indicates that the kinesin motor protein KIF13A generates tubules at the early endosomes[53]. Based on principles of geometric sorting, membrane tubulation would facilitate

separation of soluble and membrane-bound cargo[3]. Since curvature is an important determinant in regulating EHD1's membrane binding and remodeling functions, the emergent tubules could represent sites of action by EHD1. This is consistent with the estimated ~60 nm diameter of the ERC tubules[54,55], which is close to the limit for EHD1 to exert its membrane remodeling functions. On these tubes, EHD1-induced bulging could thin down the tube to facilitate fission and release of a transport carrier enriched in membrane-bound cargo. Since the pathway to fission involves a hemifusion intermediate, fission would be non-leaky thus securing the soluble cargo in the vesicular compartment. However, unlike dynamin where self-assembly and GTPase-induced conformational changes in the scaffold directly relay forces to the underlying tube causing constriction and fission[27,56], scission (at least in vitro) is a consequence of the assembly-induced tube thinning process. Such an indirect mechanism could however become effective by the formation of lipid-diffusion barriers, possibly imposed by a scaffold of ERC-resident BAR domain-containing proteins, which could restrict membrane flow and facilitate tube thinning even with limited growth of the EHD1 oligomer. Alternatively, the bulge-induced thinning of the tube could recruit an unidentified protein to catalyze fission. Both these models predict EHD1 to be necessary, but not sufficient for fission at the ERC. In this context, EHD1-binding proteins that promote or retard self-assembly could exert control over the thinning process; akin to how capping proteins control polymerization of cytoskeletal proteins and thereby

modulate force transduction. Analyzing such control mechanisms that synergize with EHD1 to release transport carriers at the ERC represents an exciting avenue for future research.

## Methods

**Constructs**. cDNAs of human EHD1 (Q9H4M9) and EHD2 (Q9NZN4) were obtained from Open Biosystems. LactC2 was a gift from Sergio Grinstein (Addgene plasmid # 22852). All constructs were cloned with an N-terminal 6xHis tag and C-terminal StrepII tag in pET15B using PCR with primers that flank the gene and the linker sequences in the vector. mEGFP was cloned either at the N- or C-terminus of EHD1 and at the N-terminus of LactC2. Site directed mutagenesis and deletions were performed using PCR. All clones were confirmed by DNA sequencing. The *vha-6* promoter was amplified from genomic DNA using the following primers (NK128 GCATGTACCTTTATAGGTGCGCTC; NK108 TTTATGGGTTTTGGT AGGTTTTAGTCGCC) and cloned in HindIII/XbaI sites of pPD49.26 vector. EHD1 and its variants were digested from pET15B using XbaI and KpnI and cloned downstream of the *vha-6* promoter in the pPD49.26 to generate the worm rescue constructs.

**Microinjections and imaging of C.elegans**. *C. elegans* were grown at 20 °C under standard conditions[57]. The strains used in this study are N2 (Bristol) wild-type line and the *rme-1(b1045)* line that has a deletion in the *C. elegans* EHD ortholog[10,20] and were acquired from the CGC stock centre. Rescue constructs (2.5 ng/ml) were injected with a co-injection marker pCFJ90 (2.5 ng/ml) and insert plasmid pBS (95 ng/ml). The injections into the gonad of *rme-1(b1045)* hermaphrodites was performed using standard procedures[58]. Basolateral uptake assays[10,20] were performed by injecting Texas-Red BSA (1 mg) into the pseudocoloem of N2, *rme-1*, and transgenic adult animals and within few minutes were imaged using a ×40/1.4NA oil-immersion lens on a Zeiss Imager Z2 equipped with an AxioCam MRm. The numbers of vacuoles were counted from multiple different lines of transgenic adult worms.

**Expression and purification of proteins**. Proteins were expressed in BL21(DE3) grown in auto-induction medium (Formedium, UK) for 24–36 h at 18 °C. Cell pellets were stored frozen at −40 °C. For purification, frozen cell pellets were thawed in 20 mM HEPES, pH 7.4, 300 mM KCl buffer and sonicated in an ice-water bath. Lysates were spun at 18,500×*g* and the supernatant was incubated with HisPur Cobalt resin (Thermo Scientific) for an hour at 4 °C. The resin was then poured into a PD-10 column, washed with 100 ml of ice-cold 20 mM HEPES, pH 7.4, 300 mM KCl buffer and eluted with 20 mM HEPES, pH 7.4, 300 mM KCl, 150 mM imidazole buffer. The elution was then applied to a 5 ml Streptactin column (GE Lifesciences) and washed with 20 mM HEPES, pH 7.4, 150 mM KCl buffer. Bound protein was eluted in 20 mM HEPES pH 7.4, 150 mM KCl buffer containing 2.5 mM desthiobiotin (Sigma). Purified proteins were spun at 100,000×*g* to remove aggregates before use in biochemical and microscopic assays.

**Lipid co-sedimentation and ATPase assay**. All lipids were purchased from Avanti Polar Lipids. Lipids were aliquoted in the required proportions into a clean glass tube, dried under high vacuum and hydrated in 20 mM HEPES, pH 7.4, 150 mM KCl buffer for 1 h at 50 °C. Liposomes containing increasing concentrations of 1,2-dioleoyl-sn-glycero-3-phospho-L-serine (DOPS) at the expense of 1,2-dioleoyl-sn-glycero-3-phosphocholine (DOPC) were extruded through 100-nm pore size filters (Avanti Polar Lipids). EHD1 (1 μM) was incubated with liposomes (100 μM) in 20 mM HEPES, pH 7.4, 150 mM KCl buffer for 20 min at room temperature. The reaction was spun at 100,000×*g* and the pellet (liposome-bound) and supernatant (free) fractions were resolved on a 10% SDS-PAGE. Gels were stained with CBB and analyzed according to ref. [16]. Protein seen in the pellet fraction with just DOPC represents non-specific binding to the tube since we see this even in the absence of liposomes. For quantation of liposome-bound fraction, we subtract this from the total. See Supplementary Figure 7 for the uncropped gel for Fig. 2a. For nucleotide hydrolysis assays, proteins (1 μM) were mixed with ATP or ATP-γ-S (Jena Biosciences, Germany) (1 mM) in the absence or presence of liposomes (100 μM) in 20 mM HEPES, pH 7.4, 150 mM KCl, 1 mM MgCl₂ and incubated at 37 °C. Aliquots were taken at regular intervals and the reaction was quenched with 5 mM EDTA. The released inorganic phosphate or thiophosphate was assayed with the malachite green reagent[59] using Na₂HPO₄ or Na₃PO₃S as standards, respectively.

**Membrane templates and fluorescence microscopy**. The membrane curvature- and protein-binding-insensitive lipid probe *p*-Texas red DHPE was separated from the mixed isomers of Texas Red DHPE (Invitrogen) using thin-layer chromatography on silica gel plates (Sigma)[28]. DOPS, DOPC, and *p*-Texas Red DHPE were reconstituted in a 40:59:1 molar ratio in chloroform and brought to a final concentration of 1 mM total lipid. Lipid stocks were stored at −40 °C and brought to room temperature before use. Membrane templates were formed by spreading 2 μl of the lipid stock on glass coverslips passivated by the covalent attachment of PEG8000[28]. The coverslip was left in vacuum and assembled in a flow cell (FCS2, Bipotechs, PA). Flowing 20 mM HEPES, pH 7.4, 150 mM KCl buffer at high flow

rates resulted in the formation of membrane tubes and a supported lipid bilayer at the source where the lipid was spotted. Buffer flow was then stopped and the tubes were allowed to settle and adhere to the surface, likely to defects in glass that resisted PEGylation. Subsequently, reactions mixtures were flowed on to these templates at tenfold slower flow rates than was used to prepare the template. For tubes with GFP bound to the inner monolayer[60], the lipid mix contained 5 mol% of the chelating lipid DGS-NTA(Ni²⁺). Templates were prepared with buffer containing 5 μM 6xHis-mEGFP as described above and once formed were stripped of mEGFP bound to the outer leaflet of the tubes by passing 100 mM EDTA. Microscopy assays were carried out at 37 °C in 20 mM HEPES, pH 7.4, 150 mM KCl buffer (assay buffer) supplemented with an oxygen scavenger cocktail[28]. Nucleotides were added to a final concentration of 1 mM to assay buffer supplemented with 1 mM MgCl₂. Templates were imaged using a ×100/1.4 NA objective on an Olympus IX-71 inverted fluorescence microscope equipped with an Evolve EMCCD (Photometrics). Confocal imaging and STED microscopy was carried out using a ×100/1.4 NA oil-immersion objective on a Leica TCS STED-3X Nanoscope using a 660 nm laser for depletion. Fluorescence recovery after photobleaching (FRAP) was carried out on templates containing 1 mol% of Rhodamine PE (RhPE, Avanti Polar Lipids) with a ×60/1.4 NA oil-immersion objective on a Carl Zeiss LSM 710 confocal microscope. Fluorescence recovery data were fitted to a model-independent empirical equation[61]. For representation purposes, $F_p$ = prebleach fluorescence, $F_b$ = fluorescence at the time of bleach, and $F_t$ = fluorescence recovering with time $t$. Dye tracing was carried out by flowing 1 μM octadecyl rhodamine B (R18, Avanti Polar Lipids) prepared in assay buffer. Template preparations present themselves with numerous membrane tubes that remain floating in solution. These were used as an assay system for observing effects on freestanding tubes.

Tube sizes were estimated according to[28] and first involved acquiring the integrated fluorescence density (ID) in ROIs of different sizes placed on background-corrected SLB images. The IDs were plotted against the ROI areas to get the slope $k_1$. Following this, IDs in ROIs of length ($l$) placed on background-corrected images of tubes were converted to their respective radii ($r$) using the formula $r = ID.(k_1.2 \cdot \pi \cdot l)^{-1}$. The maximum fluorescence intensity of tubes of various sizes were then plotted against their respective radii to get the slope $k_2$. Pixels intensities are then divided by $k_2$ to convert tube fluorescence into tube radius. Tube radius estimates reflect distance from the lumen to the center of the lipid bilayer. The combined errors in estimates arising from movement-induced fluctuations in tube fluorescence and camera noise account for <10%.

**Image and statistical analyses**. All image analysis was carried out using FIJI[62]. STED image deconvolution was carried out using Huygens. Nonlinear regression and statistical analyses were carried out using Graphpad Prism.

**Molecular dynamics simulations**. The coarse-grained force field (FF) parameters for lipid molecules are based on the hybrid coarse-graining (HCG) strategy[63], which has been extensively used to study membrane remodeling[38,64,65]. HCG integrates parameters derived from an all atom multiscale coarse graining procedure with analytical functions to augment poorly sampled regions of the configurational phase space and match the experimentally determined pair correlation, area per lipid, lateral pressure profile, and bending modulus estimates to the highly reduced membrane model[66,67]. The force field parameters for the three-site (head, body and tail) lipid models used in this work has been reported earlier[63]. The membrane composition is DOPS:DOPC (40:60 mol%) and a 100 nm-long tube is comprised of 14,500 lipids (43,500 lipid CG sites). The membrane tube chosen for this study has a length of 100 nm and an outer diameter of 20 nm and a lumen diameter of ~11 nm. Proteins are modeled as Lennard-Jones frozen sites that form a ring scaffold around the membrane tube. Such explicitly restrained protein sites have been used in simulations of membrane remodeling by protein scaffolds[68,69]. A single ring has a diameter of 25 nm and is comprised of 35 protein sites. Two adjacent sites on the ring are separated by 2.24 nm and the rings themselves are laterally apart by 0.25 nm on the scaffold. The 4 nm and 20 nm scaffolds used in this work have 560 and 2800 protein sites, respectively. The interaction between protein sites and lipid head group sites is modeled using LJ potential with $\sigma = 0.2$ nm and $\varepsilon = 1$ kcal.mol⁻¹. Several 100 nm pure membrane tubes (with no scaffolds) and lumen diameters varying from 2 to 10 nm were run for 10 million CG time steps. Tubes with very small lumen diameters (~2–4 nm) showed stochastic fission due to the forced proximity of the bilayer. Tubes with lumen diameters of 5–6 nm remained stable.

For simulations, the bilayer interacted with its periodic image in lateral dimensions. Simulations were carried out under *NVT* ensemble, using Nosé-Hoover equations of motion within the molecular dynamics suite Lammps[70]. The sizes of the box in x, y, and z dimensions were 100 nm, 50 nm, and 50 nm, respectively. The thermostat was set to 310 K, with a temperature damping parameter of 100 τ. After an equilibration period, where the time step and the temperature were slowly increased from 10 K to 310 K in 1.0 million time steps, simulations were run for 10 million integration steps of molecular dynamics simulations at a time step 1 τ. Simulations were run in three replicas per system, with different initial coordinates for lipid and protein sites. The trajectory was dumped every 10,000 τ. Each system was simulated three times by scrambling the

initial conditions. For data showing lumen radius along the tube length, mean, and variance were calculated from 20 snapshots of each simulation run.

## Data availability
The data that support the findings of this study are available from the corresponding author upon reasonable request.

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

## Acknowledgements

We thank Girish Deshpande for insightful discussions and valuable comments on the manuscript. R.D., S.C.K. and S.D. thank the Council of Scientific and Industrial Research (CSIR) and M.S.K. and N.Y.K. thank the University Grants Commission (UGC) for graduate research fellowships. K.B. and N.K. thank the Caenorhabditis Genetics Center (CGC) for the *rme-1(b1045)* strain and Yogesh Dahiya for discussions. We acknowledge supports from the IISER Pune-Leica Microscopy Imaging Centre for confocal and STED microscopy. K.B. is an Intermediate Fellow of the Department of Biotechnology-Wellcome Trust India Alliance and acknowledges funding support from the alliance [grant number IA/I/12/1/500516]. A.S. thanks the Ministry of Human Resource Development, India for a faculty startup grant and the Department of Science and Technology, India for an early career grant. A.S. acknowledges use of the high-performance computing cluster Arjun, setup from partnership grants between the Department of Biotechnology, India and IISc. T.J.P was a senior fellow of the Department of Biotechnology-Wellcome Trust India Alliance and acknowledges funding support from the alliance [grant number IA/S/16/2/502708] as well as from the Howard Hughes Medical Institute (HHMI). T.J.P. is an International Research Scholar of the HHMI.

## Author contributions

R.D., M.S.K., S.C.K., and S.D. performed biochemical and microscopic experiments. N.Y.K. performed worm experiments. N.Y.K. and K.B. designed and analyzed data from the worm experiments. A.S. performed MD simulations. T.J.P. analyzed data and wrote the manuscript.

## Additional information

**Competing interests:** The authors declare no competing interests.

