## [Peer Review File · Nature Communications]

Reviewers' comments:

Reviewer #1 (Remarks to the Author):

Overall, the authors have satisfied my main concerns.

The authors should, however, amend the graphs in Figures 2D and 3b, which incorrectly label the X-axis as "Starring tube radius" rather than "starting tube radius."

Reviewer #2 (Remarks to the Author):

With the new data, I am convinced that fission occurs on free standing tethers but the authors omitted to mention that it occurs at much shorter time scales than with the attached tethers (see movie 2 as compared to movie 1). They only show a movie that demonstrates fission but no kymograph that would clearly show the time scale difference. In fact, it seems that substrate attachment slows down the process and not favor it. This is a good point. Nevertheless, it shows that time scales are affected by the particular geometry of the assay.

Moreover, I think that although the authors have convincing data that the membrane bulges under the EHD1 oligomers and thins next to them, the mechanism leading to this membrane reshaping and eventually to fission remains unclear. Tether destabilization upon hypo-osmotic shock experiments (Movie 5 and Fig. S4) shows not only one or 2 bulges, but also very regular patterning probably corresponding to "pearls" at the limit of the resolution, which is a characteristic of pearling instabilities in membrane tethers. So far, I am not aware of other examples of tether fission through a "Rayleigh-like" instability, since these pearls are usually stable. Could the authors comment on this?

Globally, a solid physical model is still missing explaining why EHD1 induces membrane thinning next to the bulges-covers oligomers leads to membrane tether fission. The cartoon on Fig. 6 reflects only facts, supported by data, but not a mechanism. Why the membrane is thinning between the bulges has not been addressed by the experiments. Thus, Fig. 6 should only be considered as a cartoon summarizing the results of the assay, but not as a demonstration of a mechanism.

I also think that it would be interesting to discuss similarities and differences another dynamin-like protein Drp1. While Drp1 is also able to bulge membrane (see Ugarte-Urbe B. et al J. Cell Sci (2018)), it does not induce tether fission. It involves that membrane bulging and curvature gradient are not sufficient to induce fission.

To conclude, the new data presented by the authors allow to exclude some artifacts related to the experimental assay. However, they still not produce any convincing model (I mean, not a cartoon) that explains the mechanism of EHD1-mediated tether fission. I do think that this is a requirement for publishing in Nature Communications.

Reviewer #3 (Remarks to the Author):

Quantitative biophysical measurements, as described in this manuscript, are important tools to characterize protein assemblies at membranes and membrane remodeling. The authors put together a series of demanding and technically well performed experiments to characterize EHD1 assembly at membranes that, based on its scope and technical performance, could in principle warrant publication in Nature Communication. However, the experimental evidence provided in this manuscript still does not justify the main conclusions drawn by the authors.

1) The authors suggest that EHD1 is a membrane scission protein. However, as I pointed out before, there is little evidence that EHDs, in a cellular context, perform membrane scission

processes (in complete contrast to dynamin). For example, for the closely related EHD2 ATPase, just the opposite appears to be case. EHD2 localizes to the neck of caveolae, and its depletion leads to highly mobile caveolae that are budded off from the plasma membrane more easily. There is also no direct evidence so far that EHD1 is specifically recruited to the ERC and catalyzes scission of these membrane tubules; EHD1 appears to reside rather constitutively at the ERC and appears to be important for maintaining its structural integrity (and may contribute, at some point, to the fission reaction). From structural studies, it is evident that EHDs have completely different architectures and oligomerization mechanisms compared to dynamin; the proposed assembly models for EHDs, which have been supported by mutagenesis, do not indicate that EHDs are structurally suited to be efficient membrane scission molecules. However, there is ample evidence that EHDs are ATP-dependent, membrane-remodeling scaffold proteins that can create and stabilize membrane curvature, which is in complete agreement with the proposed model of the authors, as indicated.

2) The biophysical measurement described here are performed in a minimal system which is artificial in various ways: Huge EHD assemblies are formed on the provided membrane tubules (approximately 100 rings per μm), there is uncontrolled tension on these tubules, and there are no other proteins present on the membrane surface. These properties leads to breakage of the membrane tubule in the minimal setup, which is apparent in a single cleavage event per membrane tubule (indicating that membrane tension acting on both sides of the tubule has a crucial role for the scission event). Consequently, the authors suggest a model for EHD1-mediated membrane scission that may well describe the process taking place in their minimal setup. However, again as pointed out before, it is completely unclear how in a cellular context, such architecture of two oligomers growing towards each other could be accomplished; one would need to postulate two independent but coordinated oligomerization initiation events to achieve regulated membrane scission. Without a better understanding of the cellular process, the proposed model is not convincing for a cellular context.

3) The authors claim there is ATPase-dependent growth of the EHD scaffold. Again as pointed out previously, these results are difficult to reconcile with existing evidence. For example, the ATPase-deficient EHD2 variant T94A forms much larger tubules than EHD2 wt when over-expressed in cells (Daumke et al, Nature 2007). The response of the authors (e.g. a possible transition state mimicked interface) is also not convincing since EHD1 apparently undergoes many cycles of ATP hydrolysis in the analyzed time-frame. Generally, dynamin superfamily proteins form GTP/ATP dependent dimers via their GTPase domains which stabilize the scaffold and are disassembled by GTP/ATP hydrolysis or the subsequent nucleotide release. I guess that the conclusions of the authors here are blurred by minor differences in the assembly characteristics of EHD1 in the presence of ATP and the non-hydrolysable ATP γ S and AMPPNP, which is not surprising since the nucleotides are part of the assembly (G) interface. In fact, did the T94A mutant induce membrane scission of small membrane tubules in the presence of ATP and ATP γ S? How was the assembly kinetics compared to EHD2 wt in the presence of AMPPNP and ATP γ S (Fig. S2b)?

In summary, biophysical measurements on reconstituted minimal systems are important to obtain quantitative mechanistic insights into well defined cellular processes. It is, however, difficult to draw convincing evidence from minimal systems on an unknown cellular process. Therefore, instead of focusing on the EHD1-mediated scission mechanism, the authors should better concentrate in their study on new mechanistic insights in the assembly and bulging mechanism of EHD1 on membrane tubules of different sizes, including the nice and insightful comparison to dynamin (e.g. constriction mechanism versus bulging mechanism) and the curvature-dependent assembly kinetics. Such analysis could be supported by additional mutagenesis/deletion experiments, e.g. by analysis of EHD1 constructs lacking the regulatory N-terminus or the EH domain or constructs containing mutations in the G interface and stalk to obtain new mechanistic and quantitative data. One could then discuss how membrane bulges including their high membrane curvature gradients can be used for various cellular functions,

including membrane scission (as discussed for dynamin, Morlot et al, Cell 2012), stabilization of membrane curvature (as probably the case for EHD2) and also membrane fusion events (as, for example, discussed for the bacterial dynamin-like protein, Low and Lowe, Cell 2009). Of course, it should be mentioned that single membrane scission events were observed for EHD1 in the presence of ATP. However, without critically discussing the limitations of their assays and without any evidence for EHDs being membrane scission molecules in a cellular context (just the opposite for some members!), I am afraid that the study in its current form will mislead future research in the EHD field rather than support it.

Other points to the attention of the authors:

Hydrolysis of ATPgammaS by a malachite green assay.

Such experiments require calibration of the malachite green assay with thio-phosphates (e.g. not phosphates) which is non-standard. If the authors want to make a quantitative claim of reduced ATPgammaS hydrolysis by EHD1 ATPgammaS, they should properly present these results including controls in the supplement. If ATP hydrolysis was important for the scission mechanism, it is anyhow astonishing that a 10-fold reduced rate of ATP hydrolysis leads to only 2-fold reduced scission (100-fold reduced scission would be more convincing). See also comment 3 to the non-hydrolyzable ATP analogues.

Model

A similar bulging/fission model as proposed here has been previously put forward on theoretical grounds for EHD2 by Campelo, Kozlov et al. (FEBS letters, 2010), including a quantitative description of the forces required for EHD2-mediated membrane remodelling and fission (again, with the caveat that EHD2 apparently just does the opposite in a cell-based context). This is a very relevant manuscript to discuss here.

P3/4 ATP-bound EHD1-EGFP oligomers ... resembled beads on a string. This is reminiscent of how the ERC appears under conditions where EHD1 functions are inhibited.

I do not understand the rationale why inhibiting EHD1 function should lead to the same phenotype as seen by EHD1-EGFP on membrane tubules – rather the opposite is true, I would assume?

Besides, I am not sure that this is really shown in Cai et al.

Movie 2: What do we see in this movie? What are bright dots running over the tubules? Where do we see fission of free-standing tubules ?

Reviewer #1 (Remarks to the Author):

Overall, the authors have satisfied my main concerns. The authors should, however, amend the graphs in Figures 2D and 3b, which incorrectly label the X-axis as "Starring tube radius" rather than "starting tube radius."

We thank the reviewer for pointing this out and have now corrected these typos in the revised manuscript.

Reviewer #2 (Remarks to the Author):

With the new data, I am convinced that fission occurs on free standing tethers....

We thank the reviewer for being so explicit.

...but the authors omitted to mention that it occurs at much shorter time scales than with the attached tethers (see movie 2 as compared to movie 1). They only show a movie that demonstrates fission but no kymograph that would clearly show the time scale difference. In fact, it seems that substrate attachment slows down the process and not favor it. This is a good point. Nevertheless, it shows that time scales are affected by the particular geometry of the assay.

Surface attachment of the tubes could indeed have slowed down remodeling and scission. To clarify this, we have now rephrased the revised manuscript to read; "Remarkably, flowing in EHD1 with ATP caused pronounced and rapid membrane bulging and associated thinning, which led to a significant fraction of the tubes undergoing scission (Fig. 6A, red arrowheads, see Movie 1). The number of cuts on a single tube was low, possibly because the first cut led to sudden loss of tension (see Movie 1). Also, not all tubes underwent scission (Fig. 6A, white arrowheads) and a systematic analyses revealed that membrane remodeling or bulging was apparent on tubes below 25 nm in radius. Within this range, the probability of bulging leading to tube scission was sensitive to the starting tube size (Fig. 6B). Scission was also seen on freestanding tubes and at a faster rate, perhaps due to the higher membrane tension (Fig. S3A and Movie 2). However, analyzing membrane intermediates was difficult due to out-of-focus movement of the tube."

Moreover, I think that although the authors have convincing data that the membrane bulges under the EHD1 oligomers and thins next to them, the mechanism leading to this membrane reshaping and eventually to fission remains unclear. Tether destabilization upon hypo-osmotic shock experiments (Movie 5 and Fig. S4) shows not only one or 2 bulges, but also very regular patterning probably corresponding to "pearls" at the limit of the resolution, which is a characteristic of pearling instabilities in membrane tethers. So far, I am not aware of other examples of tether fission through a "Rayleigh-like" instability, since these pearls are usually stable. Could the authors comment on this?

Indeed, a pearling instability could be the cause for the hyposmotic shock-based fission process. Our experiments on templates reveal that flowing in water slowly causes bulging that quickly dissipates and the tube returns to its starting dimension. However, at faster flow rates, bulging leads to fission. Thus, a pearling instability-like mechanism can indeed cause fission provided the perturbation time scales are faster than what it takes for the pearls to heal. Since the reservoir in the tube is finite, bulging has to be compensated by thinning. Therefore, we believe these experiments inform of a mechanism by which tube bulging leads to fission.

Globally, a solid physical model is still missing explaining why EHD1 induces membrane thinning next to the bulges-covers oligomers leads to membrane tether fission. To conclude, the new data presented by the authors allow excluding some artifacts related to the experimental assay. However, they still not produce any convincing model (I mean, not a cartoon) that explains the mechanism of EHD1-mediated tether fission. I do think that this is a requirement for publishing in Nature Communications.

To further validate this pathway to fission, we now introduce new results on molecular dynamics simulations of scaffold-induced tube bulging and fission. Together, these constitute a new figure in the revised manuscript (Fig. 8). We hope this would address many of pointed concerns regarding mechanism by which a membrane bulge could lead to fission (see below for excerpts of our results section describing these results).

"Molecular dynamics simulations of scaffold-induced tube bulging and fission

We performed coarse-grained molecular dynamics simulations to gain molecular insights into how a membrane tube would behave in response to an externally applied 'attractive' scaffold that is larger in diameter than the starting tube. To mimic a short EHD1 scaffold, we placed a 4 nm-long scaffold on a 100 nm-long tube. Simulating this scenario shows that the tube under the scaffold becomes bulged (Fig. 8A). Interestingly, while the bulge extends to regions outside of the scaffold (Fig. 8B), the entire system remains stable with time (Movie 4). Performing simulations with a 20 nm-long scaffold (Fig. 8C), which should mimic the scenario reached upon adding EHD1 with ATP, produces a noticeably different outcome. The extent of tube bulging in this case is heightened which, on account of compensation of tube volume, causes progressive thinning of the tube lumen at regions outside the scaffold (Fig. 8D, Movie 5). Upon reaching a critical ~2 nm radius, lipids that are diametrically opposite in the lumen-facing leaflet are forced into close proximity, leading to spontaneous fission (Fig. 8E and Movie 5). At this stage, fluctuations in the lumen radius show a drastic increase, which suggest a possible mechanism for interleaflet mixing of lipids (Movie 6). Of note, constraints of scaffold rigidity imposed in these simulations prevent the cut ends of the tube from resealing. Indeed, simulations of a bare tube with a lumen size of ~2-4 nm, like is reached at the end of simulations with a 20 nm-long scaffold, showed stochastic fission that is followed by the resealing and separation of the cut ends of the tube (data not shown). Together, results from these simulations corroborate the fission pathway observed with EHD1 in presence of ATP. "

Inclusion of this data now makes analysis of an osmotic shock-based analysis of fission redundant and so we have removed this part from the revised manuscript. We feel the current manuscript provides data that will serve to guide future work, both from our side and from others in the community. To the best of our knowledge, this is the first study analyzing dynamics of EHD1-mediated membrane remodeling. Furthermore, in response to reviewer # 3's comments, we now include new data in the revised manuscript showing that; (a) N-terminal residues are required for stable bulging of the membrane (Fig. 9) and, (b) the closely related orthology EHD2, that forms the basis of most of our current understanding of EHD proteins, is fundamentally different from EHD1 (Fig. 10). We feel that together these make a strong case for publication of this manuscript.

The cartoon on Fig. 6 reflects only facts, supported by data, but not a mechanism. Why the membrane is thinning between the bulges has not been addressed by the experiments. Thus, Fig. 6 should only be considered as a cartoon summarizing the results of the assay, but not as a demonstration of a mechanism.

We agree and have now removed this figure and moved individual panels to other figures solely to guide the reader of our proposed mechanism.

I also think that it would be interesting to discuss similarities and differences another dynamin-like protein Drp1. While Drp1 is also able to bulge membrane (see Ugarte-Urbe B. et al J. Cell Sci (2018)), it does not induce tether fission. It involves that membrane bulging and curvature gradient are not sufficient to induce fission.

We are aware of this wonderful work. However, Ugarte-Urbe et al show that upon aspiration-induced thinning, the tube size under Drp1 scaffolds remain unaffected while adjacent regions thin down. This is very different from what we see with EHD1 where, in the absence of an externally applied change in membrane tension, EHD1 oligomers bulge the membrane. Thus, while Drp1 stabilizes curvature, EHD1 induces it. It is true that in the case of Drp1, thinning does not cause fission. However, the tubes assayed are 40-80 nm in radius, which could be well above the limit for a tube thinning process to cause fission, as we report in Fig. 6B. The paper does not report data on tubes below ~20 nm in radius, which is the range where EHD1

is able to function in fission. Additionally, in the case of EHD1, tube thinning is facilitated by the ATPase-dependent self-assembly, which is not the case for Drp1 scaffolds. We are therefore unable to relate these observations to our work.

Reviewer #3 (Remarks to the Author):

Quantitative biophysical measurements, as described in this manuscript, are important tools to characterize protein assemblies at membranes and membrane remodeling. The authors put together a series of demanding and technically well performed experiments to characterize EHD1 assembly at membranes that, based on its scope and technical performance, could in principle warrant publication in Nature Communication.

We thank the reviewer for the encouraging remarks and have tried to address as many queries as possible with new experiments, which includes analysis of the ATPase-dependence of fission, comparison to EHD2 that forms the basis for many models for how these proteins function, and for insights into mechanisms through molecular dynamics simulations that confirm our initial interpretation on the pathway to fission. We feel together these have definitely improved the scientific quality and scope of the manuscript.

However, the experimental evidence provided in this manuscript still does not justify the main conclusions drawn by the authors. The authors suggest that EHD1 is a membrane scission protein. However, as I pointed out before, there is little evidence that EHDs, in a cellular context, perform membrane scission processes (in complete contrast to dynamin).

We disagree. Previous results indicating that an absence of EHD1 leads to the expansion of ERC tubules supports the notion that it participates in membrane remodeling leading to fission at the ERC (Cai et al., MBoC 2012; Lee et al., EMBO 2015). Perhaps the complicating factor has been to regard all EHDs to work similarly, which from new results from analysis of EHD2 appears to be incorrect (see below for details).

For example, for the closely related EHD2 ATPase, just the opposite appears to be case. EHD2 localizes to the neck of caveolae, and its depletion leads to highly mobile caveolae that are budded off from the plasma membrane more easily.

Indeed, this has been puzzling for us as well. So we went ahead and analyzed EHD2 in our assays. Remarkably, new data (see Fig. 10) indicates EHD2 to be quite different from EHD1. Thus, EHD2 displays a higher affinity for PS-containing membranes but a negligible assembly-stimulated ATPase activity. Furthermore, when added to templates with ATP, EHD2 shows fission only after very long incubation times. See below for excerpts of our results and discussion section explaining these results.

"Results

Differences between EHD1 and EHD2

Previous studies indicate that liposomes incubated with EHD2 and ATP undergo tubulation with a loosely-ordered electron-dense coat on the membrane⁸, which is quite unlike the extensive membrane remodeling and scission seen here with EHD1. To test if this discrepancy reflects differences in assay conditions (preformed tubes versus liposomes), we analyzed EHD2's membrane binding, ATPase activity and membrane remodeling functions. In liposome co-sedimentation assays, EHD2 displayed a dramatically higher affinity for PS-containing liposomes, with half maximal binding seen at 10 mol% PS (Fig. 10A). Remarkably, EHD2 showed significantly lower ATPase activity than EHD1 (Fig. 10B). Thus, in presence of 100 mol% PS liposomes, EHD2 displayed an ATPase activity of 0.13 min⁻¹, which is consistent with earlier estimates⁸, but is 40-fold lower than seen with EHD1. Consistently, EHD2 showed a significant delay in membrane remodeling with ATP, with bulges appearing only after long incubation and drastically reduced scission efficiency (Fig. 10C,D). Together, our results comparing these proteins under identical experimental conditions highlight fundamental differences between the two orthologs.

Discussion

On the other hand, due to its higher affinity for negatively charged membranes and slower ATPase activity, EHD2 may function to uniformly coat the membrane rather than self-assembling at discrete sites to induce membrane curvature. Consistently, loss of EHD2 has been shown not to cause a defect in fission and release of caveolae but to enhance their mobility on the membrane ^{49,50}."

There is also no direct evidence so far that EHD1 is specifically recruited to the ERC and catalyzes scission of these membrane tubules; EHD1 appears to reside rather constitutively at the ERC and appears to be important for maintaining its structural integrity (and may contribute, at some point, to the fission reaction).

Perhaps this stems from reports that find EHD1 to be stably localized to ERC tubules, which is contrary to what is expected of a protein that catalyzes fission. Interestingly, our search of primary literature reveals that there are as many references reporting a punctate, possibly vesicular, distribution of EHD1 as there are describing its localization to membrane tubules (see list of papers below). In fact, overexpression of N-terminal GFP or epitope fusions of EHD1 display such a tubular distribution while endogenous EHD1 displays punctate localization. Based on new data shown in the revised manuscript (Fig. S1B), we think the tubular localization of N-terminal GFP fusions is an artifact since such constructs don't behave like WT EHD1 - they display no stimulation in ATPase activity with liposomes, which is a necessary attribute for fission. Due to the compromised ATPase activity, such constructs could remain stably associated to tubes and/or act in a dominant-negative manner to prevent fission by endogenous EHD1. Our work therefore cautions the use of such constructs to arrive at cellular insights into EHD1 functions.

Tubular localization			
Galperin, E., et al. (2002). Traffic 3, 575–589.	CHO cells	Overexpressed	N-terminal GFP tag, N-terminal HIV gp120 tag, fixed cells
George, M., et al (2007). BMC Cell Biol. 8, 3.	HeLa cells	Overexpressed	C-terminal GFP tagged. Fixed cells
Nasalavsky and Caplan (2005) J. Cell Sci.	HeLa cells	Overexpressed	N-terminal GFP tag. Fixed cells
Caplan, S., et al. (2002). EMBO J. 21, 2557–2567.	HeLa cells	Overexpressed	N-terminal GFP tag, N-terminal Myc-tag. Fixed cells
Jovic, M., et. al. (2007). J. Cell Sci. 120, 802–814.	HeLa cells	Overexpressed	N-terminal GFP tag
Pant, S., et al. (2009). Nat. Cell Biol. 11, 1399–1410.	HeLa cells	Overexpressed	N-terminal Myc-tag EHD1. Fixed cells
Sharma, M., et al (2010). Commun. Integr. Biol. 3, 181–183.	HeLa cells	Overexpressed	N-terminal Myc-tag EHD1. Fixed cells
Cai, B., et al. (2011). Traffic 12, 102–120.	HeLa cells	Overexpressed	N-terminal GFP-myc-tag
Bahl, K., et al. (2016) J. Biol. Chem. 291, 13465–13478.	HeLa cells	Overexpressed	N-terminal GFP-myc-tag. Fixed cells
Nasalavsky, N., et al (2004) MBOC	HeLa cells	Overexpressed	N-terminal Myc-tag. Fixed cells

Punctate localization			
Sharma, M., et al (2009) MBoC 20(24), 5181-5194.	HeLa cells	Endogenous and Overexpressed	Endogenous: EHD1 antibody, Overexpressed: myc-EHD1
Cai, B., et al (2012) MBoC 23.10: 1874-1888	HeLa cells	Endogenous and Overexpressed	Endogenous: EHD1 antibody, Overexpressed: GFP-myc-EHD1
Lasiecka, Z.M., et.al (2010).J. Neurosci. 30, 16485–16497.	Primary hippocampal neurons from day 18 rats	Overexpressed	N-terminal Flag-Tag
Guilherme, A., et al. (2004). J. Biol. Chem. 279, 40062–40075.	COS-1 cells	Overexpressed	N-terminal HA-tag. Antibody against HA-tag
Mintz, L., et al. (1999). Genomics 59, 66–76.	Adult mouse tissues, mouse embryo, COS cells	Overexpressed	N-terminal GFP tag. Fixed cells
Cai, B., Xie, et al.(2014). Front. Cell Dev. Biol. 2, 1–14.	HeLa cells	Endogeneous	EHD1-antibody
Yap, C.C.C., et al. (2010). J. Neurosci. 30, 6646–6657.	Primary hippocampal neurons from day 18 rats	Overexpressed	N-terminal Flag-Tag
Lee, D.W., et al. (2005) J. Biol. Chem. 280, 17213–17220.	HeLa cells	Overexpressed (mRme-1)	N-terminal GFP tag. Live cells
Naslavsky, N., et al (2007). Biochem. Biophys. Res. Commun. 357, 792–799.	MEFs	Overexpressed in knockout background	N-terminal Myc-tag EHD1. Fixed cells
Blume, J.J., et al. (2007). Exp. Cell Res. 313, 219–231.	NIH 3t3 cells	Endogeneous and over expressed	EHD1-antibody and HA-tagged EHD1
Mckenzie, J.E., et al. (2012). Traffic 13, 1140–1159.	BSC-1 cells	Overexpressed	N-terminal GFP- EHD1
Lin, S.X., et al. (2001). Nat. Cell Biol. 3, 567–572.	TRVb-1 cells (CHO derived)	Overexpressed (mRme-1)	N-terminal GFP tag, N-terminal Flag-tag
Lee, S., et al (2015) The EMBO journal, 34(5), 669-688.	Cos-1 cells	Endogenous	EHD1-antibody
Yeow, I., et al (2017) Current Biology, 27(19), 2951-2962	NIH 3T3 cells	Endogenous	C-terminal GFP tagged to endogenous EHD1 using CRISPR

Punctate and tubular localization			
Rotem-Yehudar, R. et al. (2001). J. Biol. Chem. 276, 33054–33060.	CHO cells	Overexpressed	EHD1-antibody. Fixed cells
Wu, C., et al. (2016). Int. J. Dev. Neurosci. 52, 24–32.	Neurons of Rat Spinal cord (1-day-old pups)	Overexpressed	N-terminal HA-tagged EHD1
Gokool, S., et al (2007). Traffic 8, 1873–1886.	HeLaM cells	Overexpressed	N-terminal GFP-tag EHD1. Fixed cells

Nevertheless, we realize that determinants that recruit EHD1 appear to be present at all times on the ERC tubules, which leads us to speculate on the mechanism by which fission is triggered. This is now discussed in the revised manuscript (see below for excerpts of our discussion section explaining these results).

"Discussion

Our work combining in vitro reconstitution and molecular dynamics simulations reveals an intrinsic ability for EHD1 to remodel and cause membrane scission. Since membranes are volume-conserving self-assemblies, bulging caused by the self-assembly of an EHD scaffold causes thinning of adjacent regions on the same tube, which results in fission. Together, these results define a novel mechanism for membrane fission. While these attributes are apparent on compositionally simple model membranes and it remains to be seen if complex native membranes, like the ERC tubules, respond similarly to EHD1, our results showing a strong correlation between membrane remodeling and fission to successful endocytic recycling strongly point to such a possibility. Importantly, previous results indicating that an absence of EHD1 leads to the expansion of ERC tubules supports the notion that it participates in membrane remodeling leading to fission at the ERC^{16,17}.

Vesicular transport pathways function to sort membrane-bound cargo and package them into transport carriers. For such a process to be effective, cargo sorting and fission of transport carriers should display a strict temporal hierarchy. Models proposed for clathrin-mediated endocytosis indicate that while the early events of cargo sorting are managed by clathrin-dependent clustering of adaptor-bound cargo, an orchestrated build-up of binding partners and an increase in membrane curvature at late stages lead to the transient

recruitment of dynamin at the necks of clathrin-coated pits, where it catalyzes fission⁵¹. On the other hand, at steady state, the ERC is abundant in lipid- and protein-binding partners of EHD1^{16,21,52}. How then can EHD1's membrane remodeling functions be timed to follow cargo sorting? Recent work indicates that the kinesin motor protein KIF13A generates tubules at the early endosomes⁵³. Based on principles of geometric sorting, tubulation facilitates the separation of soluble and membrane-bound cargo³. Since curvature is an important determinant in regulating EHD1's membrane binding and remodeling functions, the emergent tubules could represent sites of action by EHD1. This is consistent with the estimated ~60 nm size of the ERC tubules^{54,55}, which is close to the limit for EHD1 to exert its membrane remodeling functions. On these tubes, EHD1-induced bulging could thin down the tube to facilitate fission and release of a transport carrier enriched in membrane-bound cargo. Since the pathway to fission involves a hemifusion intermediate, fission would be non-leaky thus securing the soluble cargo in the vesicular compartment. However, unlike dynamin where self-assembly and GTPase-induced conformational changes in the scaffold directly relay forces to the underlying tube causing constriction and fission^{27,56}, scission (at least in vitro) is a consequence of the assembly-induced tube thinning process. Such an indirect mechanism could however become effective by the formation of lipid-diffusion barriers, possibly imposed by a scaffold of ERC-resident BAR domain-containing proteins, which could restrict membrane flow and facilitate tube thinning even with limited growth of the EHD1 oligomer. Alternatively, the bulge-induced thinning of the tube could recruit an unidentified protein to catalyze fission. Both these models predict EHD1 to be necessary but not sufficient for fission at the ERC. In this context, EHD1-binding proteins that promote or retard self-assembly could exert control over the thinning process; akin to how capping proteins control polymerization of cytoskeletal proteins and thereby modulate force transduction."

From structural studies, it is evident that EHDs have completely different architectures and oligomerization mechanisms compared to dynamin; the proposed assembly models for EHDs, which have been supported by mutagenesis, do not indicate that EHDs are structurally suited to be efficient membrane scission molecules. However, there is ample evidence that EHDs are ATP-dependent, membrane-remodeling scaffold proteins that can create and stabilize membrane curvature, which is in complete agreement with the proposed model of the authors, as indicated.

Yes and this was precisely the motivation for us to compare EHD1 with dynamin. Our results indicate that EHD1 self-assembly causes the formation of a bulge-like membrane intermediate, which is the complete opposite to what dynamin does to tubes. Remarkably however, we find EHD1 to catalyze fission of the tubes. We disagree that there's ample evidence showing that EHDs are membrane-remodeling scaffold proteins that can create and stabilize curvature. To the best of our knowledge, our manuscript is the first to comprehensively analyze membrane binding, curvature induction and scission in a dynamic set-up for any EHD protein. Also, while structural studies have informed us of how EHD2 may function, with new data presented in the manuscript (Fig. 9) we show that EHD2 is fundamentally different from EHD1 when it comes to membrane remodeling and scission.

The biophysical measurement described here are performed in a minimal system which is artificial in various ways: Huge EHD assemblies are formed on the provided membrane tubules (approximately 100 rings per μm), there is uncontrolled tension on these tubules, and there are no other proteins present on the membrane surface. These properties leads to breakage of the membrane tubule in the minimal setup, which is apparent in a single cleavage event per membrane tubule (indicating that membrane tension acting on both sides of the tubule has a crucial role for the scission event). Consequently, the authors suggest a model for EHD1-mediated membrane scission that may well describe the process taking place in their minimal setup.

This is partly true but again we believe our work informs of the biochemical and biophysical determinants that facilitate EHD1's membrane remodeling capacity. These data would constrain future models on EHD1 functions in a cellular context, and is perhaps the most significant aspect of all reconstitution studies.

Also, that uncontrolled membrane tension in these templates is a cause for scission is entirely inconsistent with what we know from membrane nanotube biophysics. For a nanotube of defined lipid composition (and bending rigidity), a change in membrane tension manifests in a change in the tube radius - higher the tension, lower is the tube radius. Since the lipid composition is constant, templates displaying tubes of varying sizes represents a system of not uncontrolled but of varying membrane tensions. Our analysis of membrane binding, remodeling and fission demonstrates a strong dependence on membrane curvature, which is the same as stating that membrane tension facilitates each of these processes. In addition, the breakage referred to by the reviewer is obviously not uncontrolled since it requires ATP hydrolysis. The infrequent cuts observed on the tubes is perhaps because the first cut results in a loss of membrane tension. We now clarify this point in the revised manuscript (see below for excerpts from the results section of the revised manuscript).

"Remarkably, flowing in EHD1 with ATP caused pronounced and rapid membrane bulging and associated thinning, which led to a significant fraction of the tubes undergoing scission (Fig. 6A, red arrowheads, see Movie 1). The number of cuts on a single tube was low, possibly because the first cut led to sudden loss of tension (see Movie 1). Also, not all tubes underwent scission (Fig. 6A, white arrowheads) and a systematic analyses revealed that membrane remodeling or bulging was apparent on tubes below 25 nm in radius. Within this range, the probability of bulging leading to tube scission was sensitive to the starting tube size (Fig. 6B). Scission was also seen on freestanding tubes and at a faster rate, perhaps due to the higher membrane tension (Fig. S3A and Movie 2)."

However, again as pointed out before, it is completely unclear how in a cellular context, such architecture of two oligomers growing towards each other could be accomplished; one would need to postulate two independent but coordinated oligomerization initiation events to achieve regulated membrane scission. Without a better understanding of the cellular process, the proposed model is not convincing for a cellular context.

This is indeed the case and is a concern for all reconstitution work with nanotubes. Since the tube is radially symmetric along its length, it is practically impossible to spatially restrict the assembly of EHD1. To circumvent this problem, we resort to molecular dynamics simulations where we test a scenario of a single scaffold that bulges the tube can cause fission. New data presented in the revised manuscript (Fig. 8) shows that this is indeed possible. Together, these results validate the proposed pathway to membrane fission (see below for excerpts of our results section explaining these results).

"Molecular dynamics simulations of scaffold-induced tube bulging and fission

We performed coarse-grained molecular dynamics simulations to gain molecular-level insights into how a membrane tube would behave in response to an externally applied 'attractive' scaffold that is larger in diameter than the starting tube. To mimic a short EHD1 scaffold, we placed a 4 nm-long scaffold on a 100 nm-long tube. Simulating this scenario shows that the tube under the scaffold becomes bulged (Fig. 8A). Interestingly, while the bulge extends to regions outside of the scaffold (Fig. 8B), the entire system remains stable with time (Movie 4). Performing simulations with a 20 nm-long scaffold (Fig. 8C), which should mimic the scenario reached upon adding EHD1 with ATP, produces a noticeably different outcome. The extent of tube bulging in this case is heightened which, on account of compensation of tube volume, causes progressive thinning of the tube lumen at regions outside the scaffold (Fig. 8D, Movie 5). Upon reaching a critical ~ 2 nm radius, lipids that are diametrically opposite in the lumen-facing leaflet are forced into close proximity, leading to spontaneous fission (Fig. 8E and Movie 5). At this stage, fluctuations in the lumen radius show a drastic increase, which suggest a possible mechanism for interleaflet mixing of lipids (Movie 6). Of note, constraints of scaffold rigidity imposed in these simulations prevent the cut ends of the tube from resealing. Indeed, simulations of a bare tube with a lumen size of $\sim 2-4$ nm, like is reached at the end of simulations with a 20 nm-long scaffold, showed stochastic fission that is followed by the resealing and separation of the cut ends of the tube (data not shown). Together, results from these simulations corroborate the fission pathway observed with EHD1 in presence of ATP."

The authors claim there is ATPase-dependent growth of the EHD scaffold. Again as pointed out previously, these results are difficult to reconcile with existing evidence. For example, the ATPase-deficient EHD2 variant T94A forms much larger tubules than EHD2 wt when over-expressed in cells (Daumke et al, Nature 2007).

Again, this could be because T94A acts as dominant-negative and interferes with EHDs fission activity. Also, as mentioned earlier, previous analysis of EHD2 in cells was carried out with N-terminal GFP fusions (Daumke et al, Nature 2007). Since N-terminal GFP tags significantly alter ATPase functions of EHD proteins (Fig. S1B), the tubular localization could have reported the localization of an ATPase-defective protein. Indeed, more recent work from Daumke's group shows that C-terminal GFP fusion of EHD2 display diffuse localization (Shah et al, Structure 2014).

The response of the authors (e.g. a possible transition state mimic-exposed interface) is also not convincing since EHD1 apparently undergoes many cycles of ATP hydrolysis in the analyzed time-frame. Generally, dynamin superfamily proteins form GTP/ATP dependent dimers via their GTPase domains which stabilize the scaffold and are disassembled by GTP/ATP hydrolysis or the subsequent nucleotide release. I guess that the conclusions of the authors here are blurred by minor differences in the assembly characteristics of EHD1 in the presence of ATP and the non-hydrolysable ATP γ S and AMPPNP, which is not surprising since the nucleotides are part of the assembly (G) interface. In fact, did the T94A mutant induce membrane scission of small membrane tubules in the presence of ATP and ATP γ S? How was the assembly kinetics compared to EHD2 wt in the presence of AMPPNP and ATP γ S (Fig. S2b)?

We agree that the differences we observe with the different nucleotides could reflect differences in the self-assembling properties of EHD1 bound to these nucleotides. To understand this further, we analyzed nucleation characteristics of EHD1 bound to AMP-PNP, ATP- γ -S and ATP and do indeed find differences (see Fig. 7A). Since assembly-stimulated ATP hydrolysis would be minimal at early stages and bulging should arise from the nucleation of EHD1 on the membrane, such differences in nucleation density could reflect different self-assembling properties of EHD1 with these nucleotides. This could explain differences in membrane binding and organization of EHD1 seen with AMP-PNP and ATP- γ -S (Fig. 2E, 4A,B). Despite these differences, ATP hydrolysis appeared to facilitate self-assembly and is evident from analysis with ATP- γ -S and the T94A mutant. Thus, reactions with ATP- γ -S, which was hydrolyzed at a 10-fold slower rate than ATP (Fig. S1A), showed a slower rate of fission (Fig. 7B). These reactions were characterized by a longer time interval between the onset of membrane bulging and tube scission (defined as the fission time, Fig. 7C). Similar effects are also seen with the T94A mutant, which self-assembled as wild type to bulge the membrane (Fig. S5A), but on account of a 10-fold slower rate of ATP hydrolysis (Fig. S1A), showed a slower rate of fission (Fig. S5B,C). Thus, slower ATP hydrolysis causes slower self-assembly which in turn causes a delay in the formation of highly thinned-down regions in between scaffolds that undergo fission.

In summary, biophysical measurements on reconstituted minimal systems are important to obtain quantitative mechanistic insights into well defined cellular processes. It is, however, difficult to draw convincing evidence from minimal systems on an unknown cellular process. Therefore, instead of focusing on the EHD1-mediated scission mechanism, the authors should better concentrate in their study on new mechanistic insights in the assembly and bulging mechanism of EHD1 on membrane tubules of different sizes, including the nice and insightful comparison to dynamin (e.g. constriction mechanism versus bulging mechanism) and the curvature-dependent assembly kinetics.

We have done so by emphasizing the membrane remodeling aspects of the manuscript and these are now supported by molecular dynamics simulations (see our response above).

Such analysis could be supported by additional mutagenesis/deletion experiments, e.g. by analysis of EHD1 constructs lacking the regulatory N-terminus or the EH domain or constructs containing mutations in the G interface and stalk to obtain new mechanistic and quantitative data.

We now provide new data in Fig. 9 addressing the role of the N-terminus in EHD1 function (see excerpts from the results section describing these results).

"Structural determinants for stable membrane remodeling

Recent studies propose that the N-terminal region of EHDs facilitate an allosteric transition from a closed auto-inhibited state in solution to an open active conformation on the membrane. Thus, while the crystal structure of full-length EHD2 reveals a compact scissor-shaped dimer where the N-terminus is tucked in the G-domain, the structure of EHD4 deleted in N-terminal residues 1-22 reveals a more open conformation⁴³. In addition, EPR spectroscopy with liposomes indicates that residues in the N-terminal region of EHD2 bind and partition into the membrane³⁶. Since the N-terminal region is partially conserved in all EHDs (Fig. 9A), we analyzed its functions in EHD1 by generating a mutant with a smaller deletion of residues 2-9, hereafter referred to as EHD1(Δ 2-9). In cross complementation assays, EHD1(Δ 2-9) was unable to rescue the *rme-1* phenotype indicating a significant defect in endocytic recycling (Fig. 9B,C). Liposome co-sedimentation assays revealed EHD1(Δ 2-9) to be similar to WT in membrane binding properties (Fig. 9D). In addition, both the basal and assembly-stimulated ATPase activities of EHD1(Δ 2-9) were similar to WT (Fig. 9E). Thus, if the N-terminal region was an allosteric regulator of ATPase activity, then EHD1(Δ 2-9) should have shown higher basal ATPase activity, which is not the case. Remarkably, reactions monitoring EHD1(Δ 2-9) on tubes with ATP showed less prominent bulging and no fission (Fig. 9F, Movie 5), again establishing that efficient endocytic recycling requires membrane remodeling and fission by EHD1. Closer analyses of reactions with ATP monitored in the membrane fluorescence channel revealed the formation but no expansion of the membrane bulge. Instead, bulges quickly dissipated and reform at the same location on the tube (Fig. 9G, white arrowheads). Together, these results indicate that the fission defect seen with EHD1(Δ 2-9) arises from instability in the scaffold that disallows sustained self-assembly in response to ATP hydrolysis."

The Δ EH domain construct of EHD1 is insoluble and so we could not analyze this mutant.

One could then discuss how membrane bulges including their high membrane curvature gradients can be used for various cellular functions, including membrane scission (as discussed for dynamin, Morlot et al, Cell 2012), stabilization of membrane curvature (as probably the case for EHD2) and also membrane fusion events (as, for example, discussed for the bacterial dynamin-like protein, Low and Lowe, Cell 2009).

We have now rephrased the discussion section to indicate such possibilities (see our response earlier).

Of course, it should be mentioned that single membrane scission events were observed for EHD1 in the presence of ATP. However, without critically discussing the limitations of their assays and without any evidence for EHDs being membrane scission molecules in a cellular context (just the opposite for some members!), I am afraid that the study in its current form will mislead future research in the EHD field rather than support it.

Please see our response above where we address these concerns.

Other points to the attention of the authors:

Hydrolysis of ATP γ S by a malachite green assay. Such experiments require calibration of the malachite green assay with thio-phosphates (e.g. not phosphates) which is non-standard. If the authors want to make a quantitative claim of reduced ATP γ S hydrolysis by EHD1 ATP γ S, they should properly present these results including controls in the supplement.

We now confirm results of ATP- γ S hydrolysis using sodium thiophosphate as a standard (see Fig.S1A).

If ATP hydrolysis was important for the scission mechanism, it is anyhow astonishing that a 10-fold reduced rate of ATP hydrolysis leads to only 2-fold reduced scission (100-fold reduced scission would be more convincing). See also comment 3 to the non-hydrolyzable ATP analogues.

New data on kinetics of fission analyzed at the level of single events indicate that fission rates, measured by estimating the time period between tube bulging and fission, is 4-fold slower with ATP- γ -S. On the other hand, a 40-fold slower ATP hydrolysis by EHD2 results in dramatically reduced fission efficiency (see Fig. 10). Admittedly, it is difficult to strictly correlate fission efficiency with rates of ATP-hydrolysis since; (a) ATPase activities are analyzed using 100 nm liposomes and we don't know the exact rates of ATP hydrolysis on membrane tubes, and (b) dimensionality could play an important role, i.e., while all EHD1 proteins hydrolyze ATP, scaffold growth leading to fission would be taking place only at the ends.

Model

A similar bulging/fission model as proposed here has been previously put forward on theoretical grounds for EHD2 by Campelo, Kozlov et al. (FEBS letters, 2010), including a quantitative description of the forces required for EHD2-mediated membrane remodelling and fission (again, with the caveat that EHD2 apparently just does the opposite in a cell-based context). This is a very relevant manuscript to discuss here.

We are aware of this work but find it difficult to reconcile this model with our data since the model posits a saddle-shaped membrane intermediate and invokes the distance of separation of EHD scaffolds to be a determinant for fission. The model does not explicitly state if EHD scaffolds are present on the saddle or abutting it. We believe our molecular dynamics simulations instead resolve this mechanism better. We also believe that this new data would provide opportunities for building better models that explicitly incorporate EHD1 assembly-induced membrane bulging as an intermediate.

P3/4 ATP-bound EHD1-EGFP oligomers ... resembled beads on a string. This is reminiscent of how the ERC appears under conditions where EHD1 functions are inhibited. I do not understand the rationale why inhibiting EHD1 function should lead to the same phenotype as seen by EHD1-EGFP on membrane tubules – rather the opposite is true, I would assume? Besides, I am not sure that this is really shown in Cai et al.

We admit this is confusing and have removed this portion in the revised manuscript.

Movie 2: What do we see in this movie? What are bright dots running over the tubules? Where do we see fission of free-standing tubules ?

The bright spots are membrane bulges that are formed due to buffer flow. This is an unavoidable consequence of working with freestanding tubes. Nevertheless, as shown in Fig. S3, stills from the movie highlight formation of membrane bulges and subsequent fission.

REVIEWERS' COMMENTS:

Reviewer #2 (Remarks to the Author):

The authors performed a very extensive revision of their manuscript and present new data, including MD simulations of the bulging/thinning process. I am convinced and agree about these changes, except for one point that has to be corrected. I think that in the process, area but not volume is conserved (the number of lipids does not change). The fluid inside the tube is free to flow in and out. In the MD simulations, when the scaffold is short, the range of membrane deformation due to the bulge is short enough compared to the total length of the tube that the rest of the tube remains unperturbed, but when the scaffold is long, the tube has to thin to globally conserve the area. In the *in vitro* experiments, the tubes are in principle connected to a reservoir (the vesicles, see Fig. 3), thus, this is not clear whether the area is really conserved or not. A possible mechanism could also be that the presence of the EHD1 scaffold on the tube reduces the diffusion of the lipids from one bare zone to another and with the reservoir, which eventually constraints the area between 2 bulges and leads to thinning. In conclusion, I would recommend to be more careful in the discussion and correct the volume conservation argument.

Reviewer #3 (Remarks to the Author):

This is a very convincing and thoughtful revision. The authors now present a strong study combining cell-based assays with biophysical experiments and molecular dynamics simulations to characterize mechanistic features of EHD1 and EHD2. Furthermore, previously published data are now consistently considered and discussed in a well balanced fashion, and the responses to my concerns are also convincing. I strongly recommend publication of the manuscript in Nature Communications.

I have a few more suggestions for text changes that the authors may want to consider. I do not need to see them again.

Line 30: Due to its significantly lower ATPase activity, the closely related EHD2 is dramatically less effective in membrane remodeling – The relation between these two functions is not entirely clear (as admitted by the authors) and the membrane remodeling/tubulating activity could be similar between the two proteins. Maybe better: The closely related EHD2 shows a significantly lower ATPase activity and, concomitantly, a dramatically decreased membrane scission activity.

Fig. 2A: One could add the information to the figure legend that neutral PC was used as second lipid. What is unclear to me: In Fig. 2A, it appears that about 50% of EHD1 is co-sedimented with PC liposomes, which may indicate 50% binding, but there is no binding quantified in Fig. 2B (same for Fig. 2D). Why is this? One could also add the liposome-free control here - does it also show 50% sedimentation, e.g. is this the reason for the quantification?

Line 96, figure 2B: Better show rates in the figures, not velocities, since rates are also reported in the text.

Line 124: N- and C-terminal EHD1-EGFP fusions – very Interesting data, especially for cell biologists. It may be worth adding a short paragraph about the possible reason and the consequences to the discussion.

Fig. 5B, legend: It would be interesting to mention how EGFP has been incorporated into the tubules, e.g. bound to the membrane or freely diffusing.

Fig. 6C/D: The figure labelling of D is slightly confusing (EHD1 + ATP). I guess both subpanels are EHD1-EGFP + ATP, and one is the protein and the other the membrane channel.

Line 209-228, Fig. 7: I am still confused about this. The effect of ATP hydrolysis on membrane scission is very clear from these experiments, but I now do not see any data showing ATPase-dependent self-assembly in the figures. Do the authors mean ATP-dependent assembly and ATPase-dependent membrane scission? In fact, that would be very interesting, since it may be a first indication that ATP hydrolysis is not only used for regulating membrane recruitment, but for an active, mechano-chemical membrane remodeling step.

Line 231: Should be Fig. 7D, not E.

Typo:

Line 41: mmechanisms

Response to Reviewers' Comments

Reviewer #2 (Remarks to the Author):

The authors performed a very extensive revision of their manuscript and present new data, including MD simulations of the bulging/thinning process. I am convinced and agree about these changes, except for one point that has to be corrected. I think that in the process, area but not volume is conserved (the number of lipids does not change). The fluid inside the tube is free to flow in and out. In the MD simulations, when the scaffold is short, the range of membrane deformation due to the bulge is short enough compared to the total length of the tube that the rest of the tube remains unperturbed, but when the scaffold is long, the tube has to thin to globally conserve the area.

We are happy that the reviewer is convinced of our proposed mechanism for fission. Yes, indeed it is the membrane area that is conserved. Our mention of volume was incorrect and has now been modified in the results and discussion sections of the manuscript (see below).

In the *in vitro* experiments, the tubes are in principle connected to a reservoir (the vesicles, see Fig. 3), thus, this is not clear whether the area is really conserved or not.

The reservoir in these templates is almost negligible after they are formed since the tubes remain connected to a supported bilayer (and not vesicles). This is apparent from our previous observations that an osmotic shock can induce membrane bulges on these tubes, which remain so for a long duration (Holkar et al., 2015, JBC). Thus, such transformation of tube morphology is a consequence of a conservation of membrane area and remains so due to the presence of the EHD1 scaffold on the bulge.

A possible mechanism could also be that the presence of the EHD1 scaffold on the tube reduces the diffusion of the lipids from one bare zone to another and with the reservoir, which eventually constraints the area between 2 bulges and leads to thinning.

Our results indicate that an EHD1 scaffold does not impose a diffusion barrier (Supplementary Figure 2). Furthermore, we also observe free diffusion of lipids under the scaffolds in our molecular dynamics simulations (not shown).

In conclusion, I would recommend to be more careful in the discussion and correct the volume conservation argument.

Per the reviewer's suggestion, we have now modified the discussion to indicate that bulge-induced tube thinning could reflect a membrane area-conserving mechanism (see below for excerpts from the revised manuscript).

"Thus, on a tube of 10 nm radius, bulging (Fig. 5G, green arrowheads) caused thinning of the intervening regions of the tube (Fig. 5G, blue lines), reaching estimates of ~7 nm in radius. Similar effects are seen with dynamin where a constriction causes bulging of adjacent regions³⁴. Such reciprocal shape change in membranes arises from the necessity to conserve membrane area, which should imply that the tubes in our templates represents a system of finite reservoir."

"Performing simulations with a 20 nm-long scaffold (Fig. 8C), which should mimic the scenario reached upon adding EHD1 with ATP, produces a noticeably different outcome. The extent of tube bulging in this case is heightened which, on account of compensation of membrane area, causes progressive thinning of the tube at regions outside the scaffold (Fig. 8D, Supplementary Movie 5)."

"Our work combining *in vitro* reconstitution and molecular dynamics simulations reveals an intrinsic ability for EHD1 to remodel and cause membrane scission. Since the total membrane

around the tube should be conserved, bulging caused by the self-assembly of an EHD scaffold causes thinning of adjacent regions on the same tube, which results in fission."

Reviewer #3 (Remarks to the Author):

This is a very convincing and thoughtful revision. The authors now present a strong study combining cell-based assays with biophysical experiments and molecular dynamics simulations to characterize mechanistic features of EHD1 and EHD2. Furthermore, previously published data are now consistently considered and discussed in a well-balanced fashion, and the responses to my concerns are also convincing. I strongly recommend publication of the manuscript in Nature Communications.

We thank the review for these positive remarks.

I have a few more suggestions for text changes that the authors may want to consider. I do not need to see them again.

Line 30: Due to its significantly lower ATPase activity, the closely related EHD2 is dramatically less effective in membrane remodeling – The relation between these two functions is not entirely clear (as admitted by the authors) and the membrane remodeling/tubulating activity could be similar between the two proteins. Maybe better: The closely related EHD2 shows a significantly lower ATPase activity and, concomitantly, a dramatically decreased membrane scission activity.

We thank the review for this suggestion and have modified the relevant portion in the revised manuscript.

Fig. 2A: One could add the information to the figure legend that neutral PC was used as second lipid.

This is now clearly stated in the methods section (see below).

What is unclear to me: In Fig. 2A, it appears that about 50% of EHD1 is co-sedimented with PC liposomes, which may indicate 50% binding, but there is no binding quantified in Fig. 2B (same for Fig. 2D). Why is this? One could also add the liposome-free control here - does it also show 50% sedimentation, e.g. is this the reason for the quantification?

This is because of non-specific sticking to the eppendorf tubes. We find the same extent of background even in assays without liposomes. For quantitation, we subtract this contribution for the total. We now clarify this in the methods.

"Liposomes containing increasing concentrations of DOPS at the expense of DOPC were extruded through 100-nm pore size filters (Avanti Polar Lipids). EHD1 (1 μ M) was incubated with liposomes (100 μ M) in 20 mM HEPES pH 7.4, 150 mM KCl buffer for 20 min at room temperature. The reaction was spun at 100,000 g and the pellet (liposome-bound) and supernatant (free) fractions were resolved on a 10% SDS-PAGE. Gels were stained with CBB and analyzed according to¹⁶. Protein seen in the pellet fraction with just DOPC represents non-specific binding to the tube since we see this even in the absence of liposomes. For quantitation of liposome-bound fraction, we subtract this from the total."

Line 96, figure 2B: Better show rates in the figures, not velocities, since rates are also reported in the text.

Both Fig. 2B and the text show the rates of ATP hydrolyzed.

Line 124: N- and C-terminal EHD1-EGFP fusions – very interesting data, especially for cell

biologists. It may be worth adding a short paragraph about the possible reason and the consequences to the discussion.

We now recommend caution to the use of N-terminal fusions in the results section.

"In contrast, N-terminal EGFP or GST fusions, used earlier to assess EHD function and distribution^{4,8,15,31}, displayed enhanced basal ATPase activity and showed no stimulation with PS liposomes (Supplementary Figure 1B). The possible reason for the N-terminal fusions not behaving like WT could be due to disruption of an important oligomerization interface (see below). We therefore recommend caution to the use of such constructs in analyzing distribution and function of EHD proteins in cells."

Fig. 5B, legend: It would be interesting to mention how EGFP has been incorporated into the tubules, e.g. bound to the membrane or freely diffusing.

We now describe this in the methods.

For tubes with GFP bound to the inner monolayer⁶⁰, the lipid mix contained 5 mol% of the chelating lipid DGS-NTA(Ni²⁺). Templates were prepared with buffer containing 5 μM 6xHis-mEGFP as described above and once formed were stripped of mEGFP bound to the outer leaflet of the tubes by passing 100 mM EDTA.

Fig. 6C/D: The figure labelling of D is slightly confusing (EHD1 + ATP). I guess both subpanels are EHD1-EGFP + ATP, and one is the protein and the other the membrane channel.

These are labeled separately since they represent different experiments. We now clarify this in the legend to Fig. 6C,D.

"**(C)** Representative montage of tube images showing the nucleation (white arrowheads) and growth of EHD1-EGFP oligomers in presence of ATP (see Supplementary Movie 3). **(D)** Representative montage of tube images showing the expansion of bulges (white arrowheads). White arrowheads mark the site of onset of bulging and green arrowheads mark sites of tube thinning."

Line 209-228, Fig. 7: I am still confused about this. The effect of ATP hydrolysis on membrane scission is very clear from these experiments, but I now do not see any data showing ATPase-dependent self-assembly in the figures. Do the authors mean ATP-dependent assembly and ATPase-dependent membrane scission? In fact, that would be very interesting, since it may be a first indication that ATP hydrolysis is not only used for regulating membrane recruitment, but for an active, mechano-chemical membrane remodeling step.

We mean that EHD1 utilizes ATP binding for membrane recruitment and ATP hydrolysis for self-assembly. The ATP-bound state (seen here with the use of the non-hydrolyzable AMP-PNP analog) forms self-limited punctae of EHD1. In contrast, reactions with ATP cause EHD1 to polymerize and cover a significant fraction of the tube. These results mean that the ATPase cycle is used in a mechanochemical membrane-remodeling step since polymerization amplifies the membrane bulge leading to thinning of the tube in intermediate regions. Thus, contrary to dynamin, which uses GTP hydrolysis to constrict tubes, EHD1 appears to use ATP hydrolysis to promote self-assembly.

Line 231: Should be Fig. 7D, not E.

Thanks. This has now been corrected.

Typo:

Line 41: mmechanisms

Thanks. This has now been corrected.